behaviour/cognition/psychology

awareness, baboon, death, mother−infant, *Papio*, thanatology

**Author for correspondence:**
Alecia J. Carter
e-mail: alecia.carter@ucl.ac.uk

# Baboon thanatology: responses of filial and non-filial group members to infants' corpses

Alecia J. Carter[1,2], Alice Baniel[3], Guy Cowlishaw[4] and Elise Huchard[1]

[1]ISEM, Université de Montpellier, CNRS, IRD, EPHE, Montpellier, France
[2]Department of Anthropology, University College London, 14 Taviton Street, WC1H 0BW, London, UK
[3]Department of Anthropology, Stony Brook University, Stony Brook, NY, USA
[4]The Institute of Zoology, Zoological Society of London, Regent's Park, London, UK

  AJC, 0000-0001-5550-9312; AB, 0000-0001-7142-5864; GC, 0000-0002-7269-4625; EH, 0000-0002-6944-449X

What do animals know of death? What can animals' responses to death tell us about the evolution of species' minds, and the origins of humans' awareness of death and dying? A recent surge in interest in comparative thanatology may provide beginnings of answers to these questions. Here, we add to the comparative thanatology literature by reporting 12 cases of group members' responses to infants' deaths, including 1 miscarriage and 2 stillbirths, recorded over 13 years in wild Namibian chacma baboons. Wild baboons' responses to dead infants were similar to other primates: in general, the mother of the infant carried the infants' corpse for varying lengths of time (less than 1 h to 10 days) and tended to groom the corpses frequently, though, as in other studies, considerable individual differences were observed. However, we have not yet observed any corpse carriage of very long duration (i.e. greater than 20 days), which, though rare, occurs in other Old World monkeys and chimpanzees. We hypothesize this is due to the costs of carrying the corpse over the greater daily distances travelled by the Tsaobis baboons. Additionally, in contrast to other case reports, we observed male friends' 'protection' of the infant corpse on three occasions. We discuss the implications of these reports for current questions in the field.

## 1. Introduction

Thanatology is the study of reactions to death and dying, and the effects of death on the surviving individuals. Comparative

thanatology aims to understand whether and how animals' responses to and understanding of death differ from humans' responses [1–3]. Little empirical research has addressed how animals respond to death. This is remarkable because understanding animals' responses to death can address fundamental questions about the evolution of cognition and emotion. For example, understanding how and why animals respond in the ways that they do to the deaths of others can shed light on questions such as: Do animals understand the difference between life and death? Can animals grieve? What are the evolutionary origins of human mortuary practices and the awareness of mortality [4]?

Most descriptions of animals' responses to death are anecdotal, and have focused on mothers' responses to the death of their infant ([3], for reviews of other cases, see [5–7]). This focus is circumstantial rather than a reflection of the putative importance of this particular behaviour for understanding responses to death: these cases are more frequently observed because of high infant mortality and the greater ability of mothers, particularly primates, to transport relatively small-bodied individuals after death [8]. Nonetheless, these records have been informative, with notable variation described in both individuals' and species' behaviour towards dead infants [9,10]. This variation has led to numerous hypotheses to explain within- and between-species variation in responses to dead infants that range from maladaptive, to neutral, to adaptive explanations ([3], reviewed for non-human primates in [4,8]). One of the most notable and best-documented behavioural responses to dead infants is the carrying of their corpses. These observations provide a good starting point for understanding the drivers of variation in individuals' responses towards dead infants. We briefly describe some of the main hypotheses for infant corpse carrying in primates below (for a more comprehensive list and review, see [8]). We focus on primates because these cases are the most commonly reported and have generated the greatest comparative thanatology literature thus far.

The most commonly cited hypotheses to explain infant corpse carrying in bereaved primates, i.e. individuals who were associated with the deceased individual before his/her death, implicate both extrinsic and intrinsic processes (reviewed in [3]). However, hypotheses involving extrinsic processes focus on environmental factors that do not explain carrying behaviour *per se* but variation in the duration of corpse carrying: corpses are carried for longer in species or populations where corpse preservation is better, i.e. dry or cold climates (the climate hypothesis: [10]). This hypothesis was not supported by a recent cross-species analysis of anthropoid primates (great apes, Old World and New World monkeys) that controlled for phylogenetic relatedness. The mother's age, the cause of the infant's death and the degree of the species' arboreality determined the length of time a corpse was carried [11].

Intrinsic factors hypothesized to influence corpse carrying include cognitive, experiential and motivational/emotional processes. In the first case, the *unawareness hypothesis* suggests that the carrier is unaware or unsure that the infant is dead, and so they persist in performing infant-care behaviours, until awareness occurs. This suggests that non-human primates lack the cognitive ability to discriminate between 'dead' and 'unresponsive but alive', and continued care is thought to be an ultimately adaptive behaviour in those cases that unresponsive individuals recover (e.g. [12]). The *infantile cues hypothesis* suggests that corpse carrying is triggered by proximate infantile cues that promote care. In the second case, experiential hypotheses implicate females' previous experience with infants. These include the *learning-to-mother hypothesis*, which suggests that nulliparous females are more likely to carry dead infants to gain mothering experience, and the *parity hypothesis*, which conversely predicts that multiparous individuals will carry infants' corpses for longer because of their greater experience as mothers [13]. In the third case, physiological and emotional states are suggested to influence corpse carrying. In particular, the *hormonal hypothesis* suggests that corpse carrying is longer in individuals with higher levels of circulating maternal hormones. This hypothesis is generally limited to explaining mothers' behaviour towards infants' corpses, or the responses of recently pregnant females. Concerning the emotional state, the *grief-management hypothesis* suggests that individuals carry the infant's corpse as a way of dealing emotionally with their loss. Finally, the *social-bonds hypothesis* was originally proposed to explain differences in bereaved individuals' responses to non-infant corpses [4], and suggests that the intensity of an individual's response to the corpse is correlated with the social bond they shared during life, and mediated by both physiological and emotional states. As mothers are most closely bonded to their infants, mothers are likely to be the most responsive.

None of these hypotheses has received consistent or unilateral support, probably because multiple factors contribute to shape individuals' responses to infants' corpses [3], several hypotheses are not mutually exclusive, as well as the dearth of long-term records of this behaviour. Overall, the current, unclear empirical record highlights the variation in reported reactions to death, and suggests the need for more data for identifying some determinants of such variation.

Here, we report 12 cases of chacma baboons' (*Papio ursinus*) responses to the pre- and postpartum deaths of infants. Our goal is to add to the growing catalogue of 'anecdotal' reports to help build a larger comparative picture of animals' responses to death. This may help to test some hypotheses regarding maternal responses to infant deaths, and ultimately to answer bigger questions on animals' mental representations of death. Following previous reports (c.f. [10], e.g. [13]), we present each case separately, providing as much detail as possible, to facilitate future comparisons within and across species. Given the recent emergence of the field of comparative thanatology, it is still unclear which aspects of these reports will prove most salient to future hypothesis testing.

# 2. Methods and results

## 2.1. Study species, site and subjects

Chacma baboons are highly social primates that live in cohesive multi-male, multi-female groups. They are omnivorous, travelling on average 1.8–7.7 km per day, with higher daily travelling distances in locations with lower rainfall and higher temperatures [14]. Chacma baboon societies are despotic with strong linear male and female hierarchies, matrilineal inheritance of rank, and promiscuous mating. Females display prominent sexual swellings during oestrus and males attempt to monopolize oestrus females in coercive 'consortships' [15]. Males can be infanticidal, with infanticide responsible for 31–37% of infant deaths in some populations [16]. Males and females may form 'friendships', in which a female associates with a particular male who is usually the father of her infant [17], through which the infant gains access to resources [18] and the female may gain protection against infanticidal males [16,19].

We studied two habituated troops (J, L) of wild chacma baboons at Tsaobis Nature Park, Namibia (15°45′ E, 22°23′ S). At the edge of the Namib Desert, Tsaobis's climate is dry (average rainfall = 85 mm [20]) and seasonal, with most rain falling between January and April [20]. The Tsaobis baboons have been under study every austral winter since 2000. Over the course of the study, troop numbers fluctuated around a median of 55 (44–69) in J troop and 52 (21–71) individuals in L troop. The baboons have little contact with humans other than the researchers, who follow them from dawn to dusk on foot. The troops forage naturally, except during research protocols including troop capture or feeding experiments. These are infrequent (five such events within the past 10 years), relatively short (two to four weeks), and generally involve provisioning the whole troop with loose maize at a single location at dawn [21,22]. Data for this study were collected opportunistically when the deaths of infants were observed during the study period. We note that the rate of infant death is much higher than reported here because the baboons are not monitored continuously, many deaths were not observed, and because not all mothers carried their offspring's corpse. In the latter case, we do not know whether the mothers quickly abandoned the corpses or whether the corpse could not be carried (e.g. due to predation or scavenging shortly after death).

## 2.2. Case reports

The 12 case reports detail troop members' responses to the deaths of infants born to 11 different mothers (one mother, GAB, is represented twice). Cases are presented chronologically. The main quantitative data from the reports are summarized in table 1, where we also include the duration before the mother resumed reproductive cycling as this has been suggested to influence carrying duration (but see [10,13]). In all cases, we give details of the infants' age at death and its sex, if known; the likely cause of death; and the parity, relative rank (top, middle or bottom third of the adult female dominance hierarchy, i.e. high, middle or low ranking, respectively) and relative age (young: less than approx. 10 years; middle-aged: approx. 10–17; older: greater than approx. 18, based on estimated ages) of the mother of the infant [3]. Where appropriate, we include details of how the corpse was carried, for how long it was carried (where known) and any other information we considered relevant. Although our sample is too small for statistical tests, we provide summary statistics of observed behaviours in table 2. Two cases concern stillbirths, three cases involve the deaths of neonates, six cases describe deaths of older dependent offspring and the last case involves the corpse of a miscarried foetus. Two cases occurred after infanticide, one a probable Bruce effect foeticide, one death presumably resulted from dehydration after the mother's death, while for the eight remaining cases the cause of the death is from known or presumed illness.

**Table 1.** A summary of the data presented in the case reports. Shown are the case numbers; the name of the case; the duration that the corpse was carried by the mother (days); the sex, age and cause of death of the infant; the rank, age and parity of the mother; and the duration before the mother was receptive (started to develop a sexual swelling).

| case no. | title | carry duration[a] | infant sex | infant age | cause of death | mother rank | mother age | mother parity | receptive[b] |
|---|---|---|---|---|---|---|---|---|---|
| 1 | RUB 2006 | 2 days* | ? | stillborn | unknown | high | young | nulliparous | 34 days |
| 2 | TRO 2006 | x | male | 0 day | unknown | low | middle | multiparous | >88 days^ |
| 3 | SUL 2006 | x | ? | 1 day | unknown | low | middle | primiparous | >80 days^ |
| 4 | GAB 2006 | 3–4 days | male | 8 months | illness | mid | young | multiparous | >29 days^ |
| 5 | MBA 2009 | 2 days* | female | stillborn | unknown | low | young | nulliparous | 142 days |
| 6 | SAL 2010 | 7 days | ? | 0 day | unknown | low | middle | multiparous | 24–37 days |
| 7 | BRA 2013 | 2 days** | male | 27–28 weeks | illness | mid | middle | multiparous | 12 days |
| 8 | PRE 2013 | 1 day§ | female | 9 months | illness | mid | middle | multiparous | 20 days |
| 9 | MYR 2013 | x | ? | miscarriage | foeticide | high | old | multiparous | 32 days |
| 10 | GAB 2014 | 10 days | male | 7 months | infanticide | mid | old | multiparous | 9 days |
| 11 | BRO 2015 | ~7 days | female | 12–13 weeks | infanticide | high | young | primiparous | 17 days |
| 12 | BIL 2017 | — | female | 9–10 weeks | dehydration | low | old | multiparous | — |

a*indicates corpses that were 'lost' by the mothers, i.e. the mothers searched for the corpse whilst emitting lost calls; **indicates that the corpse was retrieved by an observer after abandonment by the mother, but this could have limited the carrying duration; x indicates that the corpse was not carried by the mother, or was carried only briefly; §indicates the minimum carry duration (full length was not recorded); — indicates that the mother was not present to carry the corpse.

b^Note that these females did not resume cycling before the end of the field season, and the duration of receptivity may be unusually high due to a severe drought during the observation period.

**Table 2.** Summary statistics of the frequencies of observed behaviours towards infants' corpses. Shown are the behaviours and the numbers of cases, or median lengths of time that the behaviours occurred.

| behaviour | percentage (and numbers) of cases or median value |
|---|---|
| infant's corpse was carried by the mother | 81% (9 of 11) |
| infant's corpse was carried by another | 25% (3 of 12) |
| mother 'prematurely' lost the corpse (emitted lost calls on losing it) | 18% (2 of 11) |
| mother abandoned corpse on day of death | 27% (3 of 11) |
| median carrying duration if corpse was carried[a] | 3–4 days |
| median carrying duration when carried and corpse was not lost or removed[a] | 7 days |
| male showed protective behaviour towards the corpse | 25% (3 of 12) |
| individual groomed inside the corpse's mouth | 16% (2 of 12) |

[a]Does not include the case where the full carry duration was not recorded.

### 2.2.1. *Case 1:* RUB's foetus, October 2006

Stillborn foetus, carried for 2 days. Infant of unknown sex, the estimated age of foetus was *ca* 17–18 weeks based on the date of estimated conception (average gestation is 27 weeks in chacma baboons at Tsaobis). Mother high-ranking, young, nulliparous female.

#### 2.2.1.1. Case notes

The corpse was first seen hanging from RUB's vulva early in the morning, half delivered, and this situation triggered interest from several group members who stopped to check the corpse visually whilst passing by RUB. Two young juvenile males ran towards RUB to closely inspect her perineal area, to which she reacted by screaming, checking her own perineum again and running towards a thick bush to hide. The dead foetus continued to hang from RUB's vagina for about 1 h while she continued to forage. She appeared more nervous than usual. RUB was seen again 1 h later, carrying the foetus under her belly while she walked tripedally. She acted very 'nervously', and stayed near the dominant male (who consorted with her during her conceptive cycle and probably sired the foetus), similar to new mothers with live infants. After some time, she followed an old male who was often protective towards adult females and juveniles, but he did not react to her proximity. She was hyper-vigilant, and stared anxiously (through nervous glances) at any group-mate who approached. RUB later joined her mother, a middle-aged, high-ranking female, sat close to her and groomed her dead foetus while her mother groomed her own eight-month-old daughter. RUB let this young sister and her 3-year-old brother inspect the foetus. She carried the corpse ventrally for the remainder of the day, never leaving it on the ground more than a few seconds before picking it up, and walking on three legs while supporting it with her arm until climbing onto the sleeping cliff. No other group-member came to greet her during this time, which was unusual because new mothers are typically greeted very frequently on the day of birth.

   The following day RUB still carried the corpse and gave it a lot of attention, occasionally grooming it alongside her mother as she was grooming her own daughter. As the day progressed she carried it ventrally less and started to drag it. She sometimes kept it in her hand as she walked—even over rough rocky terrain—but carried it again to the sleeping cliff that evening. She generally remained close to the old protective male. The following day she still had the corpse with her, until a mid-afternoon alarm caused a panicked group movement, during which she dropped the body. Following the loss, she emitted loud calls similar but not identical to lost calls, and looked distressed. The calls continued for several hours, before progressively decreasing in frequency and finally stopping.

### 2.2.2. *Case 2:* TRO's neonate, October 2006

Neonatal infant presumably died on Day 1 of unknown cause. Not carried by mother. The infant was born approximately 10 days before the estimated term date, male. Mother low ranking, middle-aged, multiparous.

### 2.2.2.1. Case notes

The infant was born during the night or the previous day and was first observed in the first hour of troop contact (before 07.00). The infant was weak and was unable to hold onto his mother with his back legs, and was hanging underneath his mother by his arms, with his back legs hanging in the air, occasionally bumping against rocks. Occasionally TRO used an arm to support the infant and walked on three legs. The infant was not observed to reach the nipple and the mother was not observed to encourage this behaviour. She remained peripheral from the group and appeared anxious about observers' presence. By mid-morning, the infant looked very weak, and the mother left it lying on the ground while she foraged within 1 m. TRO's older daughter, 3-year-old SCA, approached the infant and held him, and he responded weakly. TRO returned upon hearing her infant's reaction, and SCA groomed TRO. TRO lay down on top of the infant. The infant made no noise, and was flattened into the sand. The mother then moved off to forage again, and again SCA returned to pick up her brother. The infant made a weak cry, and the mother came hurrying back, picked the infant up, and moved out of sight. The infant was not seen again, and no corpse carrying was recorded later that day, despite the troop being followed.

### 2.2.3. *Case 3:* SUL's neonate, November 2006

Death of unknown cause, not carried by the mother. Infant of unknown sex, 1 day old. Mother low-ranking, middle-aged, probably primiparous (no live offspring).

### 2.2.3.1. Case notes

The infant was born around term, and was first observed alive the first hour of troop contact (before 07.00), gripping its mother normally. When first observed with her infant, the mother was associating with the dominant male. However, when the mother was next observed later that afternoon (around 17.00), the baby was in the possession of a young, low-ranking and unrelated female, SCA. It is notable that this young female had lost her neonate brother 7 days earlier, and had been observed interacting with her dying brother (see previous Case Report). The mother was sitting more than 20 m away, and did not seem concerned by SCA's possession of the infant. SCA appeared agitated, moving around and aggressively protecting the infant from other inquisitive baboons. At this stage, the infant blinked once, and convulsed slightly, but otherwise looked dead. The young female continued to hold the baby, and looked very distressed when others approached. All of the baboons went onto the sleeping cliff, and the mother was not seen with her baby. The following morning, SCA continued to carry the dead infant and was very careful with it. She held the baby's face up to hers and made muzzle to muzzle contact, most likely sniffing. The level of care and attention diminished over the course of the day although she always kept the baby with her. By the afternoon SCA placed it on the ground as she foraged, and probably left the corpse on the sleeping cliff that night because she was not observed with it the following day.

### 2.2.4. *Case 4*: GAB's infant, December 2006

Death of unknown cause, though the infant showed delayed development, carried for 3–4 days. Male infant, approximately eight months old. Mother mid-ranking, young, multiparous (though no surviving offspring).

### 2.2.4.1. Case notes

The infant showed signs of locomotor and physical developmental delay: he was very small for his age, in relatively poor condition, and was carried ventrally for an unusually long period, up to five months of age. This might suggest a malformation, or a severe nutritional deficit. His mother had lost a two-month-old infant the year before. From the morning the infant was observed dead, it was carried by GAB for two full days, possibly three because there were no observers with the troop the day before the infant was recorded as dead. The mother generally carried the corpse in her mouth, occasionally dropping it to forage, and was occasionally observed grooming it. It was probably finally abandoned on a sleeping cliff as she was not observed with it the third day after the corpse was first seen.

### 2.2.5. *Case 5:* MBA's foetus, September 2009

Stillborn foetus, carried for 2 days. Female infant, almost-term foetus. Mother low-ranking, young, nulliparous female.

#### 2.2.5.1. Case notes

The probable cause of the abortion was a 2 m fall from a tree after the female had been chased there by the dominant male of her troop during an inter-troop interaction 5 days previously. On the day of the abortion, MBA was seen bleeding heavily from the vagina at 07.30. Over the following hour, she continued to bleed. She was next seen at 12.30 carrying the aborted foetus by the tail in her mouth. The corpse was still wet, suggesting that she had aborted very recently. MBA was nervous of others, including the observer, and would move away if approached by the observer or adult females. She would tolerate juveniles and infants, who showed interest in the corpse. MBA carried the corpse in her mouth by a limb, and groomed and licked the corpse when not travelling. Early on the day after the abortion, she 'lost' the corpse whilst foraging. The loss of the corpse appeared accidental—MBA searched the area in which she had been foraging making loud barks that sounded like lost calls. She continued searching as the troop moved away, and remained in the area until the troop was out of sight, before leaving to catch up with the troop. A natal adult male, 8-year-old RAB, remained near MBA while she searched the area, and travelled back to the troop with her. MBA continued to make loud barks intermittently throughout that day.

### 2.2.6. *Case 6:* SAL's neonate, June 2010

Death from unknown cause the day after birth, carried for 7 days. Infant of unknown sex, 0 days old. Mother low-ranking, middle-aged, multiparous female.

#### 2.2.6.1. Case notes

SAL carried the corpse for 7 days, frequently grooming it. Initially, she carried the corpse ventrally, but after several hours she carried it in one hand whilst travel-foraging. When travelling rapidly, she did so tripedally while holding the corpse ventrally. SAL allowed her 5-year-old daughter, SHI, to handle and groom the body, and her 3-year-old son, CYS, to play with it while SAL was being groomed by SHI 3–4 days after the infant's death. No other baboon was observed to touch the corpse, though some showed interest in it. In the days that followed, SAL started placing the corpse in her lap while she foraged, then on the ground beside her, before foraging further and further away from it. The corpse was presumably finally dropped in the night at the sleeping cliff, as SAL carried it onto the cliff in the evening but left without it the next morning.

### 2.2.7. *Case 7:* BRA's infant, June 2013

Death from probable disease, carried for 2 days before being recovered by observers. Male infant, 27–28 weeks old. Mother mid-ranking, middle-aged, multiparous female.

#### 2.2.7.1. Case notes

The infant died during the night after two weeks of deteriorating health; he had diarrhoea and defecated a large, pink nematode. Although paternity analyses have not been finalized for this infant, there were only two candidate fathers based on behavioural data (patterns of consortship during the conceptive cycle of the mother), including one whose genetic paternity can be excluded from a genotype comparison with the infant (7 mismatches across 13 microsatellite loci) (see [17] for details on genotyping methods). The second male's paternity, CHO, is highly probable based on these data (0 mismatches across 13 loci). BRA also associated with CHO during the period of the infant's death. On the morning after the infant's death, BRA foraged some distance from the corpse. CHO sat near the corpse (figure 1) and groomed it briefly at least twice. When the group started to travel rapidly, BRA and CHO remained at the rear of the troop in a seemingly agitated and confused state. BRA carried the corpse ventrally, walking tripedally and pausing regularly. She rested and barked occasionally. CHO followed and sat near BRA and the body. He was uncharacteristically wary of human observers and threatened us regularly when approached. Two observers made several attempts to recover the corpse to perform a necropsy to identify the cause of death. However, BRA screamed in protest and retrieved the corpse each time the observers approached the corpse, and

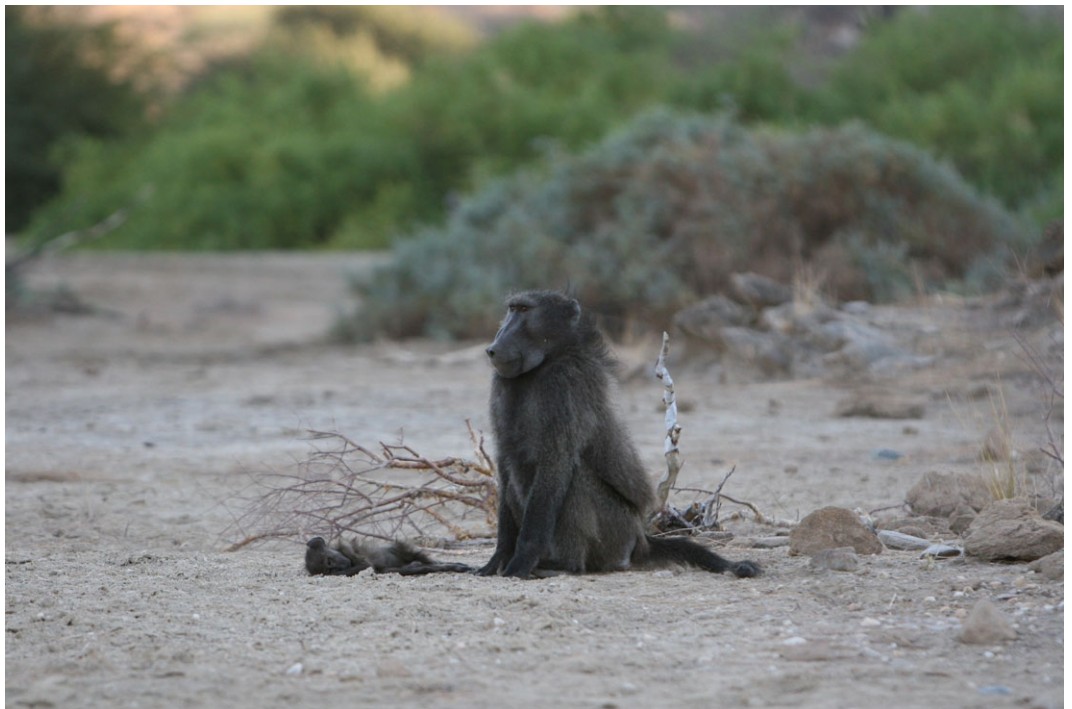

**Figure 1.** Case 7: CHO sits beside BRA's infant's corpse. Adult male CHO sits near the corpse of BRA's infant as she forages nearby. Photo by A. Carter.

CHO charged the observers several times. BRA carried the corpse for 2 days before an observer retrieved it to perform the necropsy and take a tissue sample for biopsy. At this stage, BRA was regularly foraging away from the corpse and was less vigilant towards it.

### 2.2.8. *Case 8:* PRE's infant, September 2013

Death due to illness. Female infant, nine months old. Mother middle-ranking, middle-aged, multiparous female.

#### 2.2.8.1. Case notes
Six days before the death, observers recorded a visible deterioration in the infant's condition: some yellow vomit was seen around her mouth, she was weak and lay lethargic on the ground. She was being carried ventrally by her mother, unusual for an infant of her age. The body was inspected closely by one observer the day of the death (but not retrieved) confirming no open wounds. PRE was agitated and nervous for several days following the death. She was protective of the corpse, glancing nervously if individuals paid attention to the corpse and retreating with the corpse if they approached. She groomed the corpse frequently and remained close to the dominant male VOL. PRE carried the corpse for at least 1 day.

### 2.2.9. *Case 9:* MYR's early-term foetus, September 2013

Miscarriage of early-term foetus following alpha male replacement by an immigrant male, not carried by the mother. Infant of unknown sex, foetus approximately eight weeks. Mother high-ranking, older, multiparous

#### 2.2.9.1. Case notes
MYR was pregnant with the offspring of the former dominant male, PNO, who had been the only male to consort with her during her conceptive cycle. PNO had recently lost his dominant position to LIM, who had recently immigrated into the troop. LIM had been particularly aggressive toward MYR, the dominant female, since his arrival, violently attacking her repeatedly in the weeks following his arrival. On the day of this observation, MYR was observed with vaginal bleeding. Later that day, she partially expelled a large 'clot'. MYR looked back, manually removed the foetus from her vagina, and

dropped it on the ground and walked away. LIM, following closely behind, approached the foetus and bent down to smell it for approximately 15 s. We strongly suspected a case of sexually selected foeticide. The foetus, between 10 and 15 cm long, was collected by the observer.

### 2.2.10. *Case 10:* GAB's infant, May 2014

Death probably due to infanticide, carried for 10 days. Male infant, seven months old. Mother mid-ranking, older, multiparous female.

#### 2.2.10.1. Case notes

The infant's corpse had a large cut on the right side (from which internal organs were visible); GAB also had a 10–15 cm open, bleeding cut on the right cheek. Together, these injuries were strongly suggestive of infanticide. The first day, GAB manually removed and licked the organs of the corpse that protruded from the body, but did not consume them. GAB carried the body that day, remaining at the edge of the group and aggressively protecting the corpse by screaming at other members of the group who approached. That morning, GAB left the corpse on the ground to forage briefly. At the end of the day she left the corpse for longer, but remained vigilant about approaches by other baboons or the observers. She was seen resting and in close proximity with CHO, an adult immigrant male, in the middle of the day; and with KIL, an unrelated natal male, later that afternoon. The day following the death, one observer was charged by an immigrant adult male, PLA, GAB's 'friend', as she took pictures of the corpse. PLA continued to threaten the observer until she moved more than 30 m away from the corpse. The body deteriorated rapidly in the days following the death; after 8 or 9 days, it was desiccated, with only the skin and bones remaining. GAB carried the corpse for 10 days when she started swelling again and left the body when foraging, ranging farther and farther away. However, she remained vigilant about the location of the corpse and retrieved it when an observer approached. Late on the 10th day, she was observed without the corpse.

### 2.2.11. *Case 11:* BRO's infant, June 2015

Death probably due to infanticide, carried for approximately one week. Female infant, approximately 12–13 weeks old. Mother high-ranking, young, primiparous female.

#### 2.2.11.1. Case notes

The infant died from injuries that we strongly suspect were inflicted in an infanticide: canine puncture wounds were visible in the left arm and above the hips (visible in figure 2*a*,*c*). BRO carried the corpse for approximately one week, grooming it frequently (figure 2*a*) and carrying it ventrally whilst travelling (figure 2*e*) and foraging (figure 2*b*,*c*) and after desiccation. BRO closely inspected the puncture wounds, often smelling her probing finger after doing so (figure 2*f*). She eventually pulled part of the digestive system out of one wound above the left hip by tugging on it during grooming (visible in figure 2*c*). She also cleaned debris from inside the corpse's mouth. BRO was associated with an adult male, MAR, who had consorted with BRO during the conceptive cycle for this infant (figure 2*d*). On one occasion, MAR threatened an observer when she unknowingly approached the corpse that BRO had left several metres away. BRO continued foraging nearby and was not alarmed by the observer's proximity to the subgroup or the corpse. This was not uncharacteristic, as BRO was generally relaxed around human observers. BRO allowed at least one unidentifiable juvenile female (approx. 1 year old) to approach and interact with the corpse several days after her death.

### 2.2.12. *Case 12:* BIL's infant, July 2017

Death most likely from dehydration, 2 days after the death of her mother, BIL. Female infant, approximately 9–10 weeks old. Mother low-ranking, older, multiparous.

#### 2.2.12.1. Case notes

The cause of BIL's death is unknown, but the infant, who was still dependent, survived her mother's death and remained with the troop. She was observed being carried and groomed by juvenile baboons, particularly females and an unrelated, 7-year-old male, CYS, during the 2 days she survived her mother's death. The infant attempted to forage in close proximity to these individuals. Her adult

older sister, BUB, had a similar-aged infant and did not approach or carry her sister. The infant's juvenile older sister, 4-year-old AMN, did attempt to approach her, but was often supplanted by higher-ranking individuals from different matrilines. On the afternoon of the day the infant died, the corpse was carried by CYS. This male was not previously associated closely with the infant or her mother. One observer, AC, made detailed notes on this male's and others' behaviour towards the corpse over the following day (box 1) (figures 3 and 4).

## 3. Discussion

There are few generalities to be made about chacma baboons' responses to dead infants from the 12 case reports that we present here. Although we observed some variation in the duration that mothers carried their infants' corpses, in most cases the mother carried her infant for more than 1 day (i.e. 8 of 11 cases, note that in one case (Case 12) the mother died before her infant and was, therefore, unable to carry it). However, as in other studies, there was variation in the duration (2–10 days) that the carried corpses were carried. We found no obvious influence of the mother's age or parity, the cause of the infant's death, or the length of time before the mother resumed cycling, though our sample is too small to assess these relationships statistically. In general, our anecdotes are similarly variable to those reporting many (greater than 10) cases elsewhere (i.e. [9,10]). Below, we attempt to highlight some similarities and differences between chacma baboons' responses to dead infants and those of other primates. We start by briefly assessing our observations in relation to some current hypotheses about why primates carry dead infants before considering some of the 'bigger' questions around the death of conspecifics, and what our anecdotes may suggest about the baboon mind.

The climate hypothesis suggests that prolonged carrying of infants' corpses is enabled because at high altitude or in dry climates the infant's corpse decays less rapidly (reviewed in [3,10]). Tsaobis's climate is dry and corpses desiccate rapidly, so we might expect prolonged corpse carriage. However, in contrast to reports on chimpanzees (*Pan troglodytes*) [13], gelada monkeys (*Theropithecus gelada*) [10] and Japanese macaques (*Macaca fuscata*) [9], prolonged carriage lasting weeks or months has not yet been observed. The longest carrying duration in our sample (10 days) is in line with average, not extreme, durations in other studies reporting multiple case reports from cold or dry climates [9,10]. Thus, this study does not support the climate hypothesis. However, this does not mean that climate does not play a role; other environmental factors also need to be taken into account, for example, the weight of the corpse and the average daily distance covered by the group. At Tsaobis, the average day range is approximately 6.0 km [23], making it relatively costlier to transport a corpse in comparison to the shorter day ranges of other wild species. For example, long carry durations have been reported for geladas (greater than 43 days: [10]), chimpanzees (up to 68 days: [13]), provisioned Japanese macaques (17 days: [9]) and mountain gorillas (*Gorilla beringei beringei*) (up to 27 days: [24]) and all of these species have ranges of 5 km or less: Guassa geladas (between 2.8 and 4.1 km: [25]), Bossou chimpanzees (between 3 and 5 km: S Cavalho 2020, personal communication), the provisioned Takasakiyama macaques (between 0.7 and 2.5 km: [21]) and Volcanoes National Park mountain gorillas (approx. 0.6 km: [21]). On this basis, it may be a combination of external factors that affect corpse carrying including daily travel distance interacting with climate, with wetter climates associated with more rapid decomposition of the corpse.

Do baboons realize that dead infants are dead? In general, it is difficult to determine what would constitute behavioural evidence of an awareness of death and it may be easier to identify behavioural evidence of unawareness. Some have argued that the treatment of the corpse much as if it was alive could indicate unawareness (reviewed in [3]). In support of this, most baboon mothers, and even unrelated individuals, were observed grooming the infants' corpses and carrying them ventrally. Likewise, two juveniles who interacted with BIL's infant's corpse (Case 12) 'used' it in a similar social-buffering way that live infants are used. In Barbary macaques (*Macaca sylvanus*), males of all ages have been observed to use infants' corpses in much the same way as live infants [26]. Perhaps, therefore, neither the Barbary macaques nor the baboons in Case 12 were aware that the infant was dead. Conversely, treatment of the corpse in a manner different to its treatment in life could indicate an awareness that the infant was dead (as suggested for chimpanzees: [12], see also: [3]). Our reports provide some support for this view. For example, baboons at our site have never been observed to carry live infants by dragging it in one hand along the ground, or in the mouth, even if the infant is sick or lethargic and unable to cling to the mother. Furthermore, in two cases we

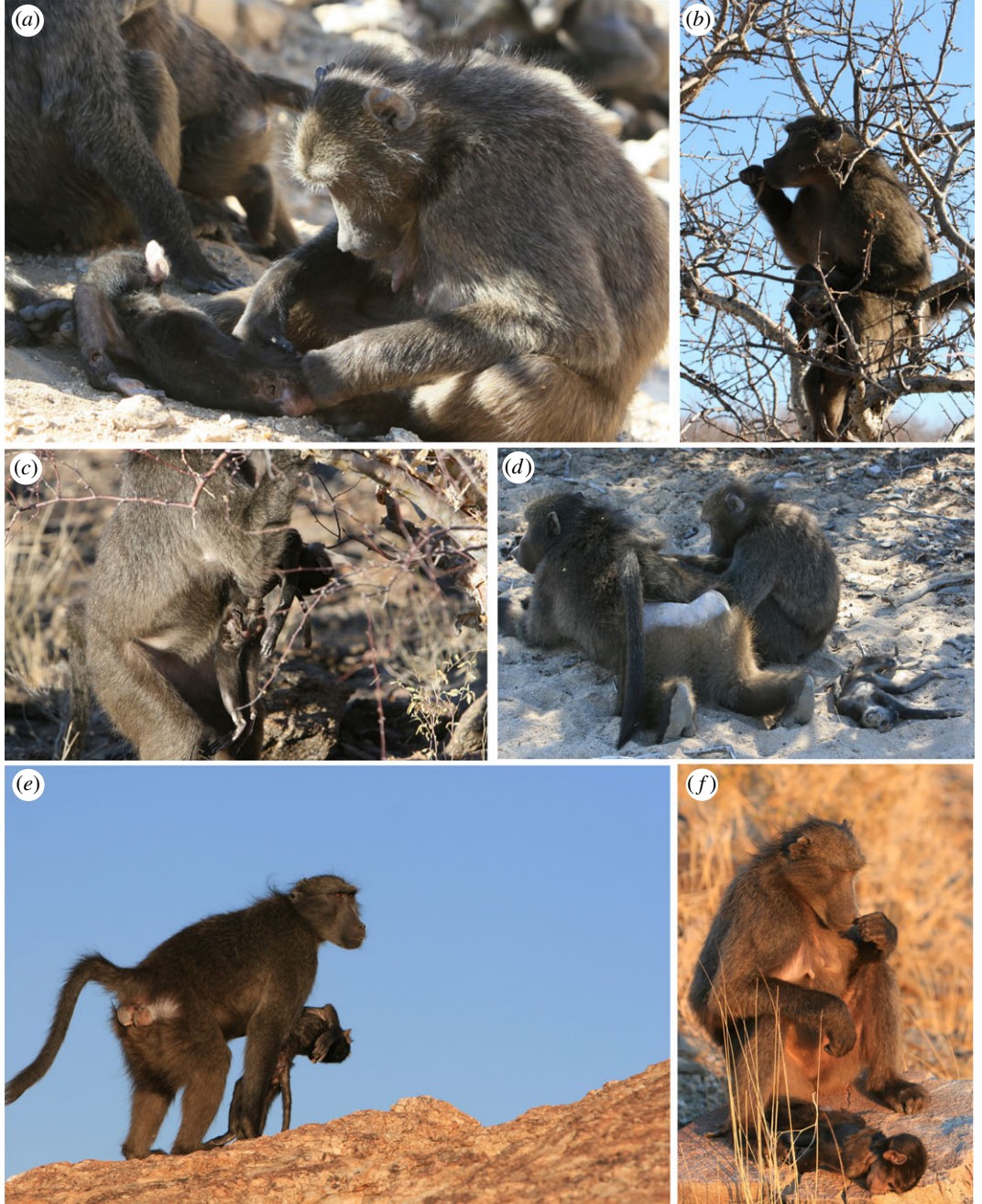

**Figure 2.** Case 11, BRO's responses to her infant's death. (*a*) BRO grooms the corpse. Note the puncture wounds visible in the arm and hip. (*b,c*) BRO forages with the corpse. Note that the digestive tract of the infant has been pulled out of the puncture wound above the hip. (*d*) BRO grooms MAR, BRO's 'friend', leaving the corpse nearby. (*e*) BRO carries the corpse ventrally. (*f*) BRO smells her finger after probing the corpse's wounds. Photos by A. Carter.

observed individuals cleaning inside the mouth of the corpse (Cases 11 and 12), which has never been observed between live individuals (discussed in greater detail below).

Does the social bond with a dead individual, either through kinship or affiliation, predict the survivor's response to the deceased? In most cases, the responding individual was the mother, who was the most closely bonded to the infant. However, in Cases 2, 3 and 12, individuals other than the mother interacted extensively with the corpse soon after death. In two of those cases, the infant died shortly after birth, the mother was still alive and the corpse was handled by a third individual, who provided greater post-mortem care than the mother. It has been argued that the lack of a strong bond if an infant dies soon after birth can result in decreased responsiveness of the mother to her infant [8]. However, our data contradict this interpretation of the social-bonds hypothesis: one mother showed extensive

carrying of her neonate's corpse (Case 6), and two stillborn foetuses were also carried for several days, with both mothers appearing distressed at losing the corpse (Cases 1 and 5). The third case where the infant's corpse was handled mostly by unrelated individuals (Case 12) lends some support to the social-bonds hypothesis: individuals observed interacting with the corpse of an already-deceased mother's infant behaved much like mothers with their own infants' corpses: they carried and groomed the body, foraged with it nearby, and protected it when others approached. However, the individuals involved performed these behaviours for less than a day whereas only mothers have been observed to carry their own infants' corpses for long periods (up to 10 days). Thus, even when given the 'opportunity' (through the absence of the mother) to carry an infant's corpse for long periods, this was not done by non-mothers in chacma baboons. Similarly, in geladas, non-mothers have been observed carrying infant corpses for a day or two but no longer [10]. However, we note that some mothers also carried their own infants' corpses for less than a day—comparable to unrelated individuals. This variability further highlights the multifactorial nature of this behaviour.

Observations of other behaviours by non-mothers lend further support to the social-bonds hypothesis. In three cases (7, 10, 11), an adult male—the friend of the mother and possible father of the dead infant in all cases—provided forms of post-mortem 'protection' involving 'guarding' behaviour by sitting near the corpse when the mother temporarily moved away, or by threatening perceived approaches to the corpse. Such guarding behaviour of an infant's corpse by an adult male has previously been reported in wild chacma baboons [27]. Notably, during one guarding event, CHO briefly groomed the infant's corpse he was guarding. Although male Barbary macaques (*Macaca sylvestris*) groom infants' corpses as they do with live infants [26], the behaviour may not be purely protective, as the infants are used as social buffers that benefit the carrying males. Infant corpse grooming is perhaps not surprising for chacma baboons given males' investment in particular live infants [17,18] and previously documented infant corpse guarding behaviour of an adult male chacma [27] and corpse carrying in an olive baboon male (*Papio anubis*) [28]. However, these behaviours appear rare in other primate species with paternal access to offspring, i.e. in other stable group-living primates with resident fathers, such as geladas, macaques and snub-nosed monkeys. Overall these observations lend further support to the social-bonds hypothesis.

We report an early-term miscarriage that was probably a 'Bruce effect'-induced abortion (Case 9) and two stillbirths (Cases 1 and 5). We note the mother's lack of response to the terminated foetus in comparison to the mothers' responses to stillborn infants. A similar observation has also been made in Yunnan snub-nosed monkeys (*Rhinopithecus bieti*), with an aborted foetus being abandoned and not carried and a stillborn foetus being carried for 1 day [29]. Although only two cases, these observations lend some support to two of the proposed hypotheses for maternal corpse-carrying behaviour. In the first case, MYR, like the unidentified snub-nosed monkey mother, may not have carried the miscarried foetus because it lacked infantile cues, supporting the *infant-cues hypothesis*. In the second case, hormones and neuropeptides hypothesized to trigger maternal carrying [30] may have been too low to elicit carrying behaviour (or any interest in the foetus) due to the early stages of the pregnancy, lending support to the *hormonal hypothesis*.

The stillbirth observations could shed light on which pregnancy or parturition hormones may be the most likely triggers of maternal carrying behaviour in very early infant deaths. For example, stillbirths are associated with lower prepartum urinary oestrones in captive hamadryas baboons (*P. anubis*) [31] and lower faecal oestrogens in the trimester of the pregnancy loss in wild yellow baboons (*Papio cynocephalus*) [32]. In both species, progesterone levels are comparable in females with live and stillbirths [31,32], suggesting that progesterone, but not oestrogen, is key in facilitating carrying behaviour after birth. This is supported by hormonal observations in low-caring and aberrant baboon mothering behaviour: hamadryas mothers engaging in less contact with their infants had higher prepartum and lower postpartum urinary metabolites of progesterone [33], suggesting that 'normal' levels of progesterone may encourage contact between a mother and her offspring, which may carry over to stillborn infants. Similarly, postpartum behavioural difficulties in hamadryas baboons were associated with elevated oestrone : progesterone ratios, again implicating (relatively) low progesterone in reduced infant care [31]. However, ovarian hormones cannot be implicated in maternal corpse-carrying behaviour of older infants because (i) ovarian hormone levels rapidly return to pre-pregnancy levels in primates, for example, in wild yellow baboons [32,34], and (ii) post-puerperium maternal behaviour is maintained not by hormones but by behavioural interactions between the mother and the infant [30]. It is possible that variation in other hormones, such as oxytocin, may be responsible for maternal corpse care post-puerperium period [28]; however, data are required to test this hypothesis.

**Box 1.** Detailed observations of troop members' responses to the corpse of an infant baboon following the loss of her mother (who died 2 days before her infant).

15 July 2017

　　1720: CYS carries the corpse in his mouth by its tail.

　　1740: CYS carries the corpse by holding it to his ventrum up and onto the sleeping cliff.

16 July 2017

　　0610: CYS carries the corpse on his ventrum; he is one of the last individuals to leave the sleeping cliff (figure 3*a*).

　　0618: CYS grooms the corpse (figure 3*b*).

　　0630: CYS grooms the corpse.

　　0645: CYS carries the corpse by the tail after the troop alarms at farmers (figure 3*c*).

　　0635: CYS approaches an adult male (unidentified) whilst carrying the corpse ventrally.

　　0649: Whilst remaining on the periphery of the group (figure 3*d*), CYS grooms the corpse, not foraging (figure 3*e*).

　　0656: CYS grooms the corpse, then carries the corpse very 'gently' by the tail.

　　0721: CYS starts foraging whilst carrying the corpse; he is on the periphery of the troop with RUB, a high-ranking adult female.

　　0741: CYS receives grunts from NEC, a 7-year-old juvenile male, who approaches; CYS responds by tail-flagging and retreating, carrying the corpse by the tail.

　　0754: CYS transports the corpse by dragging it by the tail along the ground.

　　0805: Whilst foraging in a large *Salvadora persica* bush, CYS threatens ASP, a 5-year-old juvenile male, who approached the corpse without having seen it, as CYS had left it on the ground outside the bush while he foraged in the bush.

　　0805: CYS threatens ANA, CYS's 4-year-old female sister, who approached the corpse. CYS then returns to the ground to gather the corpse, and carries it into the Salvadora where he forages with it in his lap (figure 3*f*).

　　0901: CYS carries the corpse by its right hand in his mouth. The troop alarms several times; each time CYS grooms the infant after the alarm, in a similar fashion to the grooming elicited between pairs after a troop alarm.

　　0926: CYS climbs into a *Prosopis glandulosa* tree with the infant by holding it to his ventrum and in his mouth by the tail.

　　1000: CYS threatens ULC, a higher-ranking, 7-year-old, adult female, unrelated to either CYS or the infant, for access to the corpse.

　　1005: CYS grooms the corpse.

　　1025: CYS threatens AMN, the 4-year-old sister of the dead infant, for approaching the corpse, where he had left it on the ground approximately 8 m away. He takes the corpse into the tree.

　　1054: BRU, an unrelated 7-year-old subadult male, approaches the corpse; CYS picks up the corpse to approach BRU whilst lip-smacking and making a come-hither face, an affiliative gesture (figure 3*g*). BRU returns the gestures and looks at the corpse.

　　1109: CYS grooms the corpse.

　　1143: The corpse is found on the ground by ANA and groomed, and she cleans its mouth, removing some debris and inspecting it (figure 4*a,b*). ANA is approached by CYS, but not supplanted, and by AMN.

　　1146: AMN approaches the corpse and grooms it, but is displaced and chased by Narco, a 6-year-old, high-ranking male; ANA relocates and grooms the corpse. CYS returns but does not approach ANA.

　　1205: CYS forages nearby.

　　1218: ANA approaches BUB, the 9-year-old adult female sister of the dead infant, with the corpse in a manner similar to that of juvenile females 'returning' infants to mothers (figure 4*c*). BUB does not respond to ANA's approach.

　　1220: ANA climbs a tree tripedally with the corpse.

　　1230: ANA is approached by BOT, an unrelated, 2-year-old juvenile female, who then briefly carries the corpse.

　　1300: End of detailed observations.

　　Approximately 1400: ANA drops the corpse during a troop alarm and it is not subsequently recovered.

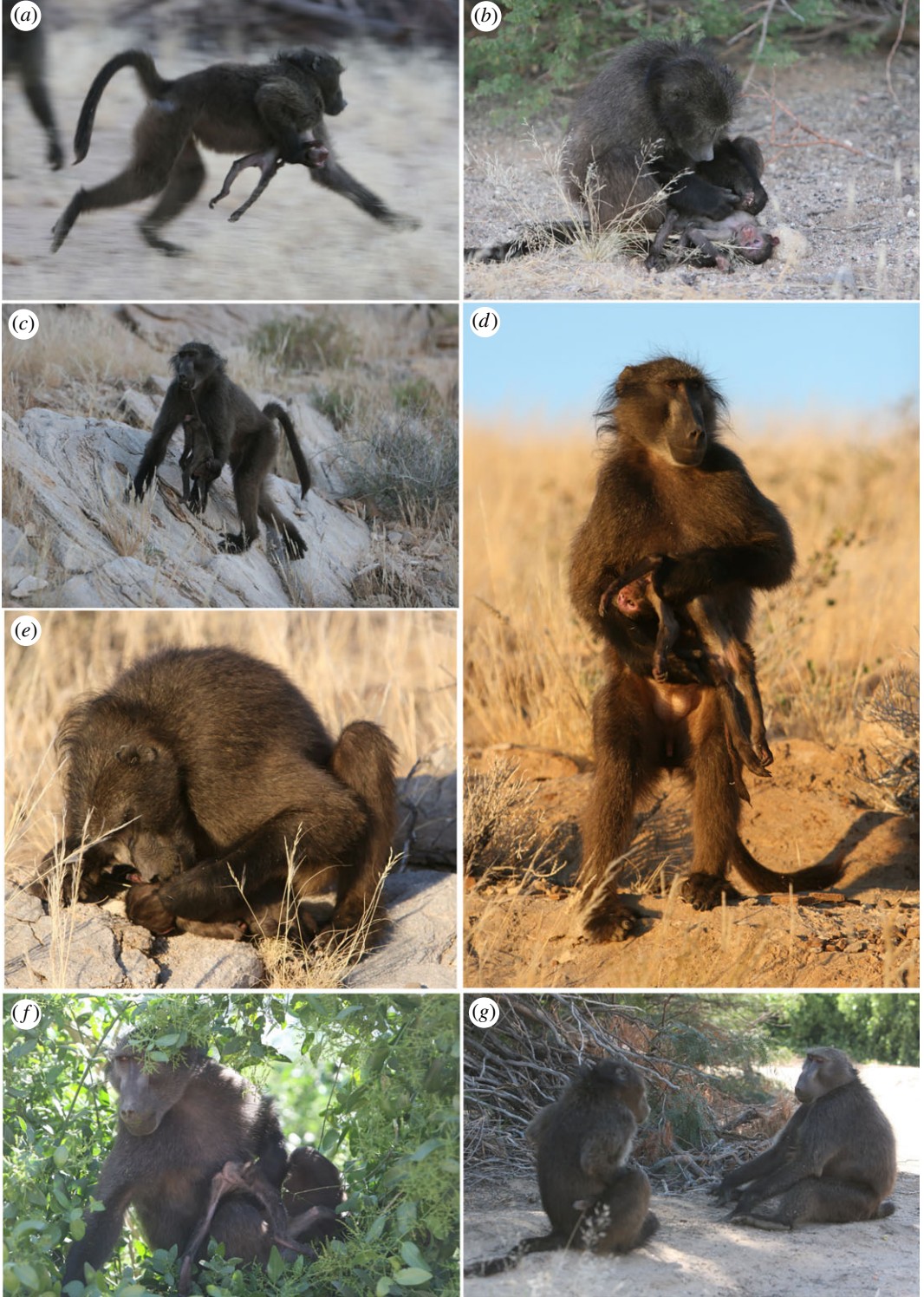

**Figure 3.** Case 12, CYS's responses to BIL's dead infant. (*a*) CYS carries the corpse ventrally from the sleeping cliff. (*b*) CYS grooms the corpse on leaving the sleeping cliff. (*c*) CYS carries the corpse in his mouth by the tail. (*d*) CYS remains on the periphery of the troop, not foraging and continuing to handle the corpse. (*e*) CYS continues to groom the corpse on the periphery of the troop. (*f*) CYS forages with the corpse after it was approached by other troop members. (*g*) Using the corpse in as a social buffer, CYS approaches BRU, who performs a lip-smack at the corpse. Photos by A. Carter.

Finally, we report a rare thanatological behaviour, namely cleaning the mouth and wounds of the deceased (Cases 11 and 12). Two individuals were observed cleaning inside the mouth of two different infant corpses, removing and inspecting debris. One of these two individuals also cleaned a wound on her infant's corpse. Similar behaviours have been recently reported in a captive Tonkean

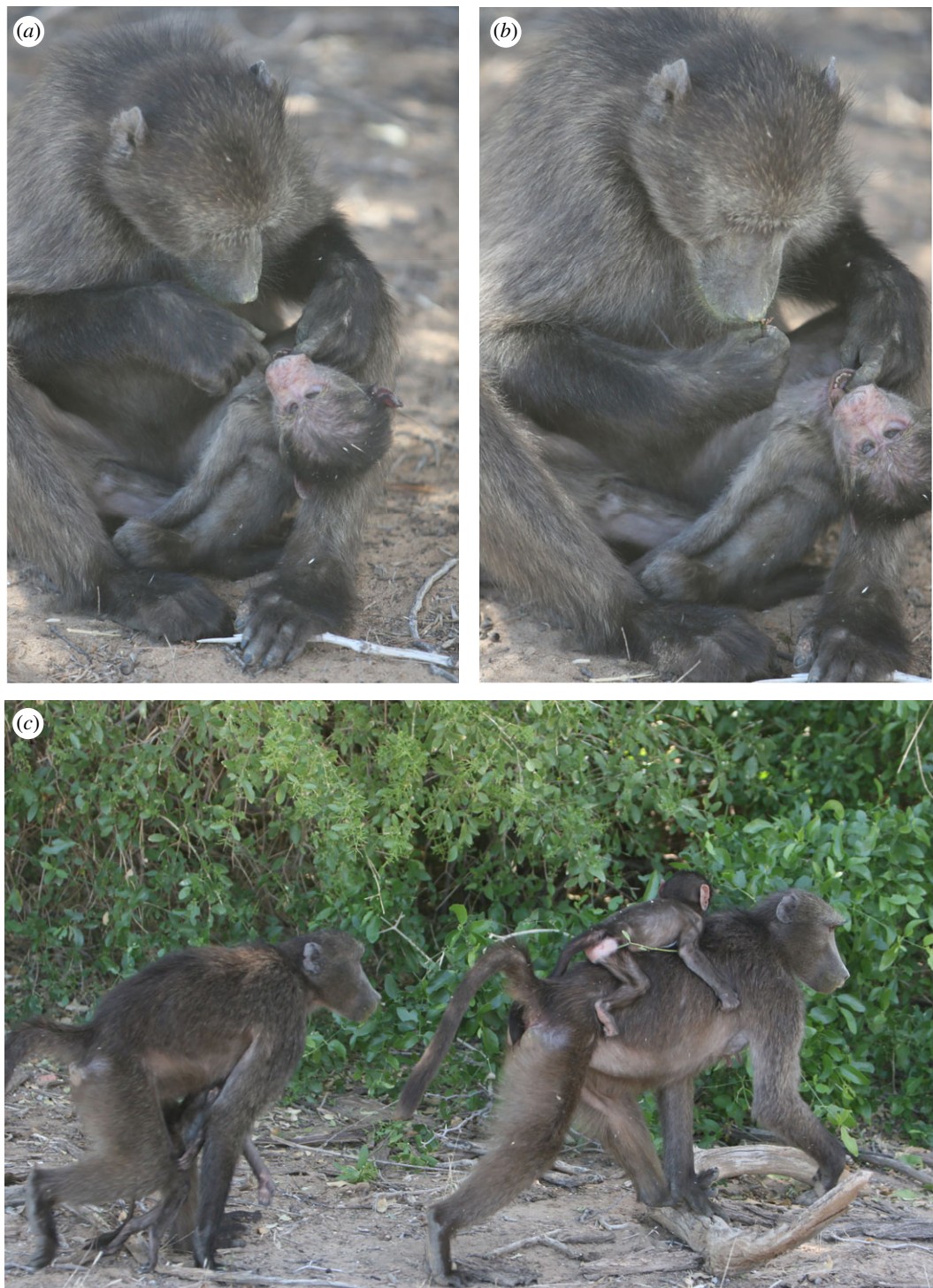

**Figure 4.** Case 12, ANA's responses to BIL's dead infant. (*a*) ANA cleans inside and (*b*) removes debris from the mouth of the corpse. (*c*) ANA approaches BUB, BIL's adult daughter, who had a similar-aged son at the time of BIL's infant's death. Photos by A. Carter.

macaque mother (*Macaca tonkeana*) [35] and a juvenile female bonnet monkey (*Macaca radiata*), unrelated to the dead infant whose mouth she investigated [11]. Similar observations have given rise to discussions about compassion towards the dead individual, as was the case for cleaning the teeth of an adolescent male chimpanzee's corpse [7]. Evidence for compassion has also been hypothesized in an adult male Sichuan snub-nosed monkey's (*Rhinopithecus roxellana*) behaviour towards a recently deceased individual, which was described as 'exploration, attempts to elicit a response from the (deceased individual), and affiliative acts including embracing' [36, p. R404].

Does mouth- and wound-cleaning behaviour, or affiliative acts towards a corpse, suggest 'compassion' towards dead individuals? We call for caution, for two reasons. First, no clear definition of compassion has been proposed by these previous studies, but compassion in humans is defined as 'the feeling that arises in witnessing another's suffering and that motivates a subsequent desire to help' [37, p. 351]. Compassion cannot be directly observed and must be inferred indirectly through behavioural manifestations. In humans, non-verbal compassion can be quantified through facial expressions, body orientation and touch [37]. Perhaps, then, the described behaviours are evidence of compassion, but the provided justifications are tenuous. In the first case, the justification was that the mouth-cleaning behaviour was 'prolonged and detailed' [7, p. 921] (the behaviour lasted approx. 2 min); in both cases, the authors cite the fact that the behaviour was performed by an individual who had a close social bond to the deceased individual as evidence of compassion [7,36]. While the perception of an experienced observer is valuable when trying to interpret a complex behaviour, it is unclear why such behaviour would be more compelling than a long grooming bout directed toward a different body location, for example, which is frequent in the primate thanatological literature (reviewed in [3]). The duration of such behaviour could simply reflect the fact that the corpse is unresponsive. In our study, affiliative acts (grooming) were performed by most individuals who interacted with the corpses, not only those that were closely bonded, and an unrelated, unbonded individual was observed cleaning the mouth of a dead infant, indicating that this unusual behaviour is not restricted to closely bonded individuals.

Second, no alternative explanations to compassion have been proposed to explain these atypical behaviours, which could, for instance, equally indicate grief management responses in the bereaved individuals, or curiosity about death in the non-bereaved individuals. What is clear is that teeth- and mouth-cleaning behaviours have never been observed being directed towards live baboons at our field site, and that in itself makes them valuable to describe and discuss. (We note, however, that teeth-cleaning with tools has been described in chimpanzees [38].) Empirical attempts to decipher the nature of the mental and emotional processes underlying such observations will undoubtedly be very challenging, and may be best addressed by establishing systematic data collection protocols around dead and seriously injured individuals in order to support some level of hypothesis testing, as well as to document the apparent emotional state of individuals involved in contacts with corpses.

We add new cases to the growing body of literature on primate thanatology, and show how new data can generate both support—or lack of support—for existing hypotheses and more questions to address. Although quantitative treatments of responses towards infants' corpses should be targeted to move the field forward [8], we hope that we have demonstrated that systematic publication of 'detailed anecdotes' (see also: [39]) can also further our understanding of some of the 'big' thanatological questions, especially as new reports accumulate from a variety of sites and species. We, therefore, encourage future data presentation in this detailed qualitative format as well as, if possible, implementation of specific behavioural data protocols related to encounters with death.

Ethics. Our research procedures were evaluated and approved by the Ethics Committee of the Zoological Society of London and the Ministry of Environment and Tourism, Namibia (MET Research/Collecting Permits 1039/2006, 1379/2009, 1486/2010, 1786/2013, 2009/2015, 2303/2017), and adhered to the ASAB/ABS Guidelines for the Treatment of Animals in Behavioural Research and Teaching.

Data accessibility. All relevant raw data are published in table 1 of the manuscript, and qualifying data are contained in the associated case reports. These data are complete and are not aggregated.

Authors' contributions. A.J.C., A.B., G.C. and E.H. collected field data and critically revised the manuscript; A.J.C. and E.H. conceived the study. A.J.C. coordinated the study and drafted the manuscript. All authors gave final approval for publication and agree to be held accountable for the work performed therein.

Competing interests. At the time of writing, Dr Alecia Carter is a Board Member of Royal Society Open Science, but had no involvement in the review or assessment of the paper.

Funding. The authors are grateful to the following funders who supported their research at Tsaobis: the Leakey Foundation (2009–2010; 2016–2017), the Animal Behavior Society (USA) (2009), the International Primatological Society (2009), the Explorers Club Exploration Fund (2009); Primate Society of Great Britain (2013); the Gibbs Travelling Research Fellowship, Newnham College (2015); Fondation des Treilles (2017); and Agence Nationale de la Recherche Labex IAST (2017).

Acknowledgements. We thank the Tsaobis Baboon Project field teams from 2006 to 2017 for their dedication and patience every year that made following the baboons possible. We thank Harry Marshall for passing on the details of MBA's day 1 response. We thank Susana Carvalho for providing unpublished data. We are grateful to the Ministry of Lands and Resettlement for permission to work at Tsaobis Leopard Park, the Gobabeb Training and Research Centre for affiliation, and the Ministry of Environment and Tourism for research permission in Namibia. We are also grateful to the Snyman and Wittreich families for permission to work on neighbouring farms. This paper is a publication of the ZSL Institute of Zoology's Tsaobis Baboon Project. ISEM contribution number 2020-036.

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
