## [Reviewer comments · Royal Society Open Science]

Review History

RSOS-192206.R0 (Original submission)

Review form: Reviewer 1

Is the manuscript scientifically sound in its present form?

Yes

Are the interpretations and conclusions justified by the results?

Yes

Is the language acceptable?

Yes

Do you have any ethical concerns with this paper?

No

Have you any concerns about statistical analyses in this paper?

No

Recommendation?

Accept with minor revision (please list in comments)

Comments to the Author(s)

As the authors state, it is important for more animal behaviour researchers to describe and publish carefully observed and described instances of responses to dead conspecifics, to advance the field of evolutionary/comparative thanatology. Here, the focus is on responses by mothers and other individuals to dead infant chacma baboons, a species that appears under-represented in the literature. The authors provide several nice descriptions of what happened to dead infants of varying ages and likely causes of death, and relate their observations to some hypotheses that exist in the literature. I will be happy to see this paper published.

I have no major criticisms of the work, but shortly after I started to read the manuscript I felt that it was over-written, with too many redundant phrases, repetitions, and a lack of concision. I'm attaching the manuscript (Appendix A) with many suggested changes aimed at increasing conciseness and making the paper more readable to a scientific, rather than a popular readership. Some of the comments are questions, so the authors are advised to read them all carefully and take it from there.

Additionally, I offer some further comments aimed at making the paper both tighter and stronger:

The recently published paper by Das et al. deserves more than one passing citation, as it also addresses several hypotheses using a relatively large and multi-species data base.

The authors repeatedly surmise that an attending adult male was the likely or probably father of the dead infant. But without supporting evidence this is too speculative. The baboons are studied every winter, but doesn't that mean that the females conceived at a time when no observations were conducted, and therefore the father is simply unknown? If so, I would recommend dropping the suggestion each time, and somehow working into the Discussion a single sentence alluding to the fact that some of the attending adult males appeared to have long-term friendships with the mothers.

Case 5: I'd like more information. When was the foetus expelled after the mother's fall?

In several places the authors describe "protection" of the corpse when others approached. Again, some clarification would be helpful.... Was protection ever aggressive, or did the corpse-handler simply avoid approaches by others, for example moving away, or simply shielding the corpse with its body?

Line 474: When discussing day ranges of species in relation to length of corpse-carrying, I'd recommend trying to get day range length information for the specific study sites where long durations have been reported, e.g., Bossou for chimpanzees.

Line 619: I totally disagree that mouth-grooming and teeth-cleaning have "never" been observed between living individuals. I've watched stumptail macaques and tonkean macaques fold back the lips of a groomee and carefully groom the exposed area, and I suspect that somewhere in the hundreds of papers and chapters on grooming that this kind of behaviour is described (albeit not highlighted as being special; it may be simply included as face-directed grooming). Bill McGrew also published a paper entitled "chimpanzee dentistry," in which the interactants were very much alive! Again, mouth-grooming in chimps might well be much more common than the authors suggest.

I hope these comments and suggestions are useful.

Review form: Reviewer 2

Is the manuscript scientifically sound in its present form?

Yes

Are the interpretations and conclusions justified by the results?

Yes

Is the language acceptable?

Yes

Do you have any ethical concerns with this paper?

No

Have you any concerns about statistical analyses in this paper?

No

Recommendation?

Accept with minor revision (please list in comments)

Comments to the Author(s)

I think this article provides important implications to expand our understanding on the animals' responses to death. Existing literatures on primate thanatology have mainly come from forested areas with humid climate. This article presents unique data-set from a dried habitat of chacma baboons that travel long distances and use sleeping cliff at night. The comparison with other primates will highlight the characteristics in each clade; even in a dry habitat, the length of carrying in chacma baboons was not so much different from other monkey species.

Points to be considered in the revise is listed below.

P3 L11: The word "grieve" is too speculative. I suggest to remove this sentence from the abstract. Instead, I advise to include the description of "awareness" since the word is included in the key words but missing from the abstract text.

P6 L92: Put parenthesis to the species name, chacma baboons' (*Papio ursinus*) responses

P6 L104: More descriptions on the study site and the subject troops should be included in this part. The climate is estimated to be dry, but authors should give concrete information such as annual rainfall and seasonality, if any (and add month to each case report). The composition of the two study troops should be included, even if it's rough summary, such as xx-xx individuals for troop J (including x-x adult male and x-x adult female). Also, give basic information for behavioral characteristics of chacma baboons for readers other than primatologists; diet, daily travel distance, society, mating (and consort?).

P6 L112: higher than the (previous?) reports suggests... The following long sentence was not so clear for me. Inclusion of the total number of observation days during the 13 years from 2000 would be informative to have rough ideas about the frequency of observing infants' deaths.

P7 L131-133: Number of cases for each category should be checked; $2+3+7+1=13$

P9 L187: When was the timing of confirming the mother without the infant? Some hours later on the same day or the next morning?

P10 L217: lost a two-month-old infant the year before (not included in this report due to the lack of precise records?)

P10 L227: MBA carried the corpse in which manner (ventrally or in hand/mouth)?

P12 L279: Why the length of carrying was not recorded in this case? Even if the end day was not recorded, you may say that it was carried at least for x days.

P13 L286: Please explain the possible cause of infanticide and frequency of happening in chacma baboons.

P13: Case 9 and Case 10 happened in the similar timing?

P13 L303: please add the note of "not carried (by the mother)"

P16-18: Box 1 may be better to be included as a supplementary material, leaving some important parts (such as social buffering between males in L397-) in the main text.

P27: The part discussing about "compassion" may be too speculative. Please try to discuss your gained results in a scientific manner.

P28 L605: p. 921 of which reference?

Decision letter (RSOS-192206.R0)

21-Jan-2020

Dear Dr Carter,

On behalf of the Editors, I am pleased to inform you that your Manuscript RSOS-192206 entitled "Baboon thanatology: Responses of filial and non-filial group members to infants' corpses" has been accepted for publication in Royal Society Open Science subject to minor revision in accordance with the referee suggestions. Please find the referees' comments at the end of this email.

The reviewers and handling editors have recommended publication, but also suggest some minor revisions to your manuscript. Therefore, I invite you to respond to the comments and revise your manuscript.

- Ethics statement

- Data accessibility

<http://datadryad.org/submit?journalID=RSOS&manu=RSOS-192206>

- Competing interests

- Authors' contributions

- Acknowledgements

- Funding statement

Because the schedule for publication is very tight, it is a condition of publication that you submit the revised version of your manuscript before 30-Jan-2020. Please note that the revision deadline will expire at 00.00am on this date. If you do not think you will be able to meet this date please let me know immediately.

- 1) A text file of the manuscript (tex, txt, rtf, docx or doc), references, tables (including captions) and figure captions. Do not upload a PDF as your "Main Document";
- 2) A separate electronic file of each figure (EPS or print-quality PDF preferred (either format should be produced directly from original creation package), or original software format);
- 3) Included a 100 word media summary of your paper when requested at submission. Please ensure you have entered correct contact details (email, institution and telephone) in your user account;

- 4) Included the raw data to support the claims made in your paper. You can either include your data as electronic supplementary material or upload to a repository and include the relevant doi within your manuscript. Make sure it is clear in your data accessibility statement how the data can be accessed;
- 5) All supplementary materials accompanying an accepted article will be treated as in their final form. Note that the Royal Society will neither edit nor typeset supplementary material and it will be hosted as provided. Please ensure that the supplementary material includes the paper details where possible (authors, article title, journal name).

If your manuscript is newly submitted and subsequently accepted for publication, you will be asked to pay the article processing charge, unless you request a waiver and this is approved by Royal Society Publishing. You can find out more about the charges at <https://royalsocietypublishing.org/rsos/charges>. Should you have any queries, please contact opscience@royalsociety.org.

Best regards,
Lianne Parkhouse
Editorial Coordinator
Royal Society Open Science
opscience@royalsociety.org

on behalf of Dr Atsushi Iriki (Associate Editor) and Kevin Padian (Subject Editor)
opscience@royalsociety.org

Reviewer comments to Author:

Reviewer: 1
Comments to the Author(s)

As the authors state, it is important for more animal behaviour researchers to describe and publish carefully observed and described instances of responses to dead conspecifics, to advance the field of evolutionary/comparative thanatology. Here, the focus is on responses by mothers and other individuals to dead infant chacma baboons, a species that appears under-represented in the literature. The authors provide several nice descriptions of what happened to dead infants of varying ages and likely causes of death, and relate their observations to some hypotheses that exist in the literature. I will be happy to see this paper published.

I have no major criticisms of the work, but shortly after I started to read the manuscript I felt that

it was over-written, with too many redundant phrases, repetitions, and a lack of concision. I'm attaching the manuscript with many suggested changes aimed at increasing conciseness and making the paper more readable to a scientific, rather than a popular readership. Some of the comments are questions, so the authors are advised to read them all carefully and take it from there.

Additionally, I offer some further comments aimed at making the paper both tighter and stronger:

The recently published paper by Das et al. deserves more than one passing citation, as it also addresses several hypotheses using a relatively large and multi-species data base.

The authors repeatedly surmise that an attending adult male was the likely or probably father of the dead infant. But without supporting evidence this is too speculative. The baboons are studied every winter, but doesn't that mean that the females conceived at a time when no observations were conducted, and therefore the father is simply unknown? If so, I would recommend dropping the suggestion each time, and somehow working into the Discussion a single sentence alluding to the fact that some of the attending adult males appeared to have long-term friendships with the mothers.

Case 5: I'd like more information. When was the foetus expelled after the mother's fall?

In several places the authors describe "protection" of the corpse when others approached. Again, some clarification would be helpful.... Was protection ever aggressive, or did the corpse-handler simply avoid approaches by others, for example moving away, or simply shielding the corpse with its body?

Line 474: When discussing day ranges of species in relation to length of corpse-carrying, I'd recommend trying to get day range length information for the specific study sites where long durations have been reported, e.g., Bossou for chimpanzees.

Line 619: I totally disagree that mouth-grooming and teeth-cleaning have "never" been observed between living individuals. I've watched stumptail macaques and tonkean macaques fold back the lips of a groomee and carefully groom the exposed area, and I suspect that somewhere in the hundreds of papers and chapters on grooming that this kind of behaviour is described (albeit not highlighted as being special; it may be simply included as face-directed grooming). Bill McGrew also published a paper entitled "chimpanzee dentistry," in which the interactants were very much alive! Again, mouth-grooming in chimps might well be much more common than the authors suggest.

I hope these comments and suggestions are useful.

Reviewer: 2

Comments to the Author(s)

I think this article provides important implications to expand our understanding on the animals' responses to death. Existing literatures on primate thanatology have mainly come from forested areas with humid climate. This article presents unique data-set from a dried habitat of chacma baboons that travel long distances and use sleeping cliff at night. The comparison with other primates will highlight the characteristics in each clade; even in a dry habitat, the length of carrying in chacma baboons was not so much different from other monkey species.

Points to be considered in the revise is listed below.

P3 L11: The word “grieve” is too speculative. I suggest to remove this sentence from the abstract. Instead, I advise to include the description of “awareness” since the word is included in the key words but missing from the abstract text.

P6 L92: Put parenthesis to the species name, chacma baboons’ (*Papio ursinus*) responses

P6 L104-: More descriptions on the study site and the subject troops should be included in this part. The climate is estimated to be dry, but authors should give concrete information such as annual rainfall and seasonality, if any (and add month to each case report). The composition of the two study troops should be included, even if it’s rough summary, such as xx-xx individuals for troop J (including x-x adult male and x-x adult female). Also, give basic information for behavioral characteristics of chacma baboons for readers other than primatologists; diet, daily travel distance, society, mating (and consort?).

P6 L112: higher than the (previous?) reports suggests... The following long sentence was not so clear for me. Inclusion of the total number of observation days during the 13 years from 2000 would be informative to have rough ideas about the frequency of observing infants’ deaths.

P7 L131-133: Number of cases for each category should be checked; $2+3+7+1=13$

P9 L187: When was the timing of confirming the mother without the infant? Some hours later on the same day or the next morning?

P10 L217: lost a two-month-old infant the year before (not included in this report due to the lack of precise records?)

P10 L227: MBA carried the corpse in which manner (ventrally or in hand/mouth)?

P12 L279: Why the length of carrying was not recorded in this case? Even if the end day was not recorded, you may say that it was carried at least for x days.

P13 L286: Please explain the possible cause of infanticide and frequency of happening in chacma baboons.

P13: Case 9 and Case 10 happened in the similar timing?

P13 L303: please add the note of “not carried (by the mother)”

P16-18: Box 1 may be better to be included as a supplementary material, leaving some important parts (such as social buffering between males in L397-) in the main text.

P27: The part discussing about “compassion” may be too speculative. Please try to discuss your gained results in a scientific manner.

P28 L605: p. 921 of which reference?

Author's Response to Decision Letter for (RSOS-192206.R0)

See Appendix B.

Decision letter (RSOS-192206.R1)

18-Feb-2020

Dear Dr Carter,

It is a pleasure to accept your manuscript entitled "Baboon thanatology: Responses of filial and non-filial group members to infants’ corpses" in its current form for publication in Royal Society Open Science.

on behalf of Dr Atsushi Iriki (Associate Editor) and Professor Kevin Padian (Subject Editor)
openscience@royalsociety.org

Appendix A**ROYAL SOCIETY
OPEN SCIENCE****Baboon thanatology: Responses of filial and non-filial group members to infants' corpses**

Journal:	Royal Society Open Science
Manuscript ID	RSOS-192206
Article Type:	Research
Date Submitted by the Author:	19-Dec-2019
Complete List of Authors:	Carter, Alecia; University College London, Department of Anthropology Baniel, Alice; Stony Brook University, Department of Anthropology Cowlshaw, Guy; Institute of Zoology, ; Huchard, Elise; Institut des sciences de l'evolution, Biologie Evolutive Humaine
Subject:	behaviour < BIOLOGY, cognition < BIOLOGY, psychology < BIOLOGY
Keywords:	baboon, death, mother-infant, thanatology, Papio, awareness
Subject Category:	Biology (whole organism)

Author-supplied statements

Relevant information will appear here if provided.

Ethics

Does your article include research that required ethical approval or permits?:

Yes

Statement (if applicable):

Our research procedures were evaluated and approved by the Ethics Committee of the Zoological Society of London and the Ministry of Environment and Tourism, Namibia (MET Research/Collecting Permits 1039/2006, 1379/2009, 1486/2010, 1786/2013, 2009/2015, 2303/2017), and adhered to the ASAB/ABS Guidelines for the Treatment of Animals in Behavioural Research and Teaching.

Data

It is a condition of publication that data, code and materials supporting your paper are made publicly available. Does your paper present new data?:

Yes

Statement (if applicable):

All relevant raw data are published in Table 1 of the manuscript, and qualifying data are contained in the associated case reports. These data are complete and are not aggregated.

Conflict of interest

I/We declare a competing interest

Statement (if applicable):

At the time of writing, Dr Alecia Carter is a Board Member of Royal Society Open Science, but had no involvement in the review or assessment of the paper.

Authors' contributions

This paper has multiple authors and our individual contributions were as below

Statement (if applicable):

AC, AB, GC and EH collected field data and critically revised the manuscript; AC and EH conceived the study. AC coordinated the study and drafted the manuscript. All authors gave final approval for publication and agree to be held accountable for the work performed therein.

Baboon thanatology: Responses of filial and non-filial group members to infants' corpses

Alecia J. Carter^{1,2,*}, Alice Baniel³, Guy Cowlshaw⁴ & Elise Huchard¹

¹ISEM, Université de Montpellier, CNRS, IRD, EPHE, Montpellier, France

²Department of Anthropology, University College London, UK

³Department of Anthropology, Stony Brook University, Stony Brook, NY, USA

⁴The Institute of Zoology, Zoological Society of London, Regent's Park, London, UK

*Author for correspondence: Department of Anthropology, University College London, 14 Taviton St, WC1H 0BW, London, UK. E: alecia.carter@ucl.ac.uk

What do animals know of death? Do animals grieve? What can animals' responses to death tell us
about the evolution of species' minds, and the origins of human's understanding of death and dying?
A recent surge in interest in comparative thanatology may provide beginnings of answers to these
questions. Here, we add to the comparative thanatology literature by reporting 12 cases of group
members' responses to infants' deaths, including 1 miscarriage and 2 stillbirths, recorded over 13
16 years in wild Namibian chacma baboons. Wild baboons' responses to dead infants were similar to
17 other primates: in general, the mother of the infant carried the infants' corpse for varying lengths of
18 time (<1 hour-10 days) and tended to groom the corpses frequently, though, as in other studies, high
variation was observed between different individuals' responses to the infants' corpses. However,
we have not yet observed any corpse carriage of very long duration (i.e. >20 days), which, though
rare, occurs in other old world monkeys and chimpanzees. We hypothesise this is due to the costs of
carrying the corpse over the greater daily distances travelled by the Tsaobis baboons. Additionally,
in contrast to other case reports, we observed paternal "protection" of the infant corpse on two
occasions. Our case reports add to the growing literature on primate thanatology, and we discuss the
implications of these reports for current questions in the field.

Key words: awareness; baboon; death; mother-infant; *Papio*; thanatology

INTRODUCTION

Thanatology is the study of reactions to death and dying, and its effect on the surviving individuals.
Comparative thanatology aims to understand whether and how animals' responses to and
understanding of death differ from humans' responses [1–3]. Little empirical research has addressed
how animals respond to death. This is remarkable because understanding animals' responses to
death can address fundamental questions about the evolution of cognition and emotion. For
example, understanding how and why animals respond ~~in the ways that they do~~ to the deaths of
others can shed light on ~~the~~ question ~~Do animals understand the difference between life and~~
death? Can animals grieve? ~~And,~~ what are the evolutionary origins of human mortuary practices and
the awareness of mortality [4].

Most descriptions of animals' responses to death are anecdotal, and have focussed on
mothers' responses to the death of their infant. [3. For reviews of other cases, see 5–7] This focus is
circumstantial rather than a reflection of the putative importance of this particular behaviour for
understanding responses to death: these cases are more frequently observed because of high infant
mortality and the greater ability of mothers, particularly primates, to transport relatively small-
bodied individuals after death [8]. ~~Despite this accidental focus,~~ these records have been informative,
with notable variation described in both individuals' and species' behaviour towards dead infants
[9,10]. This variation has led to ~~the generation of~~ numerous hypotheses to explain within- and
between-species variation in responses to dead infants that range from maladaptive, to neutral, to
adaptive explanations [3,reviewed for non-human primates in 4,8]. One of the most notable and
best-documented behavioural responses to dead infants is the carrying of their corpses. These
observations provide a good starting point for understanding the drivers of variation in individuals'
responses towards dead infants. We briefly describe some of the main hypotheses for infant-corpse
carrying in primates below [for a more comprehensive list, ~~including a review of supporting literature~~
see 8]. We focus on primates because these cases are the most commonly reported and have
generated the greatest ~~animal~~ thanatology literature thus far.

The most commonly-cited hypotheses to explain infant-corpse carrying behaviour in primates
implicate ~~processes that are both extrinsic and intrinsic to bereaved individuals,~~ i.e. individuals who
were associated with the deceased individual before ~~his/her~~ death [reviewed in 3]. However, ~~the~~
hypotheses involving extrinsic processes focus on environmental factors that do not explain carrying

behaviour *per se* but variation in the duration of corpse-carrying: corpses are hypothesised to be carried for longer in species or populations where corpse preservation is better i.e. dry or cold climates [the climate hypothesis: 10].

The intrinsic factors that are hypothesised to influence corpse-carrying behaviour include cognitive, experiential and state-based processes. In the first case, the *unawareness hypothesis* suggests that the individual carrying the corpse is unaware or unsure that the infant is dead, and that they persist in performing infant-care behaviours, until this awareness arises. This suggests that non-human primates lack the cognitive ability to discriminate state “dead” from “unresponsive but alive,” and is thought to be an ultimately adaptive behaviour in those cases that unresponsive individuals recover [e.g. 11]. The *infantile cues hypothesis* suggests that corpse carrying behaviour is promoted by proximately-selected cognitive cues that promote infant care. In the second case, experiential hypotheses implicate females’ previous experience with infants. These include the *learning-to-mother hypothesis*, which suggests that nulliparous females would be more likely to carry dead infants to gain mothering experience, and the *parity hypothesis*, which conversely predicts that multiparous individuals will carry infants’ corpses for longer because of their greater experience as mothers [12]. In the third case, physiological and emotional states are suggested to influence corpse-carrying behaviour. In particular, the *hormonal hypothesis* suggests that corpse carrying behaviour is promoted by an individual’s hormonal state, with longer carrying occurring in individuals with higher levels of circulating hormones that are implicated in promoting maternal behaviour. This hypothesis is generally limited to explaining mothers’ behaviour towards infants’ corpses, or the behaviour of individuals who are or have recently been pregnant. At the emotional state level, the *grief-management hypothesis* suggests that individuals carry the infants’ corpses during the time required emotionally to deal with their loss. The role of both physiological and emotional states could be influenced by the social bond shared by the deceased infant and the bereaved individual. The *social-bonds hypothesis* was originally proposed to explain differences in bereaved individuals’ responses to non-infant corpses [4], and suggests that the intensity of an individual’s response to the deceased’s corpse is correlated with the social bond they shared during life. As mothers are most closely bonded to their infants, mothers could be predicted to be the most responsive.

None of these hypotheses has received consistent or unilateral support, which probably reflects the fact that multiple factors contribute to individuals’ responses to infants’ corpses [3], that several hypotheses are not mutually exclusive, as well as the dearth of long-term records of this

behaviour. Overall, the state of the current empirical record, with nearly as many hypotheses as anecdotal observations, highlights the variation of reported reactions to death, and suggests that many more data will be required to start identifying some determinants of such variation.

Here, we report twelve cases of chacma baboons' *Papio ursinus* responses to the deaths of pre- and postpartum infants. Our goal is to add to the growing catalogue of "anecdotal" reports that describe primates' responses to death to contribute to build a larger comparative picture of animals' responses to death. This may help to test some hypotheses regarding maternal responses to infant deaths, and ultimately to answer bigger questions on animals' mental representations of death. Following previous case reports [c.f. 10, e.g. 13], we present each case separately, providing as much detail as possible, in the hope that this format will be most useful for future comparisons within and between species. Given the recent emergence of the field of comparative thanatology, it is still unclear which aspects of these reports will prove most salient to future hypothesis testing.

METHODS AND RESULTS

Study site and subjects

We studied two habituated troops (J, L) of wild chacma baboons at Tsaobis Nature Park, Namibia (15°45'E, 22°23'S). The Tsaobis baboons have been under study every austral winter since 2000. The baboons have little contact with humans other than the researchers, who follow them from dawn to dusk on foot. The troops forage naturally, except during particular research protocols, including troop capture, or feeding experiments. These are uncommon (five such events within the past 10 years), relatively short (2–4 weeks), and generally involve provisioning the whole troop with loose maize at a single location at dawn [14,15]. Data for this study were collected opportunistically when the deaths of infants were observed during the study period. We note that the rate of infant death is much higher than the reports suggest; this is because the baboons are not monitored continuously and many deaths were not observed, and because not all mothers carried their offspring's corpse. In the latter case, however, it is not known whether the mothers abandoned the corpses upon their death or whether the corpse was able to be carried (e.g. due to predation or scavenging shortly after death).

Case reports

~~Below we present 12 case reports detailing the behaviour of~~ troop members' responses to the deaths
 of infants born to ~~eleven~~ different mothers (one female, GAB, is represented twice). Cases are
 presented chronologically. The main quantitative data from the reports are summarised in Table 1,
 where we also include the duration before the mother resumed ~~g~~ling as this has been suggested
 to influence carrying duration [but see 10,12]. In all cases, we give details of the infants' age at death
 and its sex, if known; the likely cause of death; and the parity, relative rank (~~whether they were in~~
 ~~the~~ top, middle or bottom third of the adult female dominance hierarchy, i.e. high, middle or low
 ranking, respectively) and relative age (young: <~10 years; middle-aged: ~10-17; older: >~18, based
 on estimated ages) of the mother of the infant [3]. Where appropriate, we include details of the
 ~~method that the individuals involved used to carry the corpse, the duration that the corpse was~~
 carried (~~when known~~) and any other ~~observations of interest for improving our understanding of~~
 ~~what animals know about and how they respond to death~~. Although our sample is too small to
 ~~provide~~ statistical tests, we ~~do~~ provide summary statistics of the frequencies of observed behaviours
 in Table 2. Two cases ~~describe responses to the corpses of stillbirths~~, three cases describe the deaths
 of neonates, seven cases describe ~~mother~~ dependent offspring, and the last case ~~describes the~~
 ~~response to~~ the corpse of a miscarried foetus. Two cases ~~describe death~~ after infanticide, one a
 probable Bruce effect foeticide, one death ~~from~~ presumed dehydration after the mother's death,
 whilst for the remaining eight the cause of the death is from known or presumed illness.

**Case 1: RUB's foetus, 2006:** Stillborn foetus, carried for 2 days

*Infant sex and age:* Unknown sex, the estimated age of foetus was ca. 17-18 weeks based on the date
 of estimated conception (average gestation is 27 weeks in chacma baboons at Tsaobis)

*Maternal rank, age and parity:* High-ranking, young, nulliparous female.

*Case notes:* The corpse was first seen hanging from RUB's vulva early in the morning, half delivered,
 and this situation triggered interest from several group members. ~~They appeared intrigued and~~
 stopped to check the corpse visually whilst passing by RUB. Two young juvenile males ran towards
 her for a detailed inspection of her perineal area, to which she reacted by screaming, checking her
 own perineum again, and running towards a thick bush to hide. The ~~foetus' body remained partially~~
 ~~delivered, hanging from her~~ vagina for about one hour while RUB continued to forage. She appeared
 more nervous than usual. RUB was seen again one hour later, carrying the foetus under her belly,
 walking tripodally. She acted very "nervously," and stayed in the proximity of the dominant male

(who consorted with her during her conceptive cycle and likely sired the foetus), in a similar manner
to that of new mothers with live infants. After some time, she followed an old resident male who
regularly acted very protectively towards females and juveniles, was hyper-vigilant, and stared
anxiously at any group-mate who approached. RUB later joined her mother, a middle-aged, high-
150 ranking female, and sat very close to her, grooming her dead foetus while her mother was grooming
her own 8-month old daughter. RUB let this young sister and her 3 y. o. brother inspect her foetus.
She carried her infant ventrally for the remainder of the day, never leaving it more than a few seconds
on the ground before picking it up, and walking on three legs while supporting it with her arm until
climbing onto the sleeping cliff. Surprisingly, no other group mate came around her to greet her
during that time period, which contrasts with the usual attitude of group members towards a new
mother, who is greeted very frequently on the day of birth.

The following day RUB still carried the corpse and gave it a lot of attention, occasionally
grooming it alongside her mother, who groomed her own daughter. As the day progressed she
carried it ventrally less and started to drag it. She sometimes kept it in her hand as she walked—even
when walking over rough rocky terrain—but carried it again to the sleeping cliff that evening. She
spent a lot of time in the proximity of the old protective male. The following day she still had her
infant with her, until an alarm occurred mid afternoon of that day, followed by a panicked group
movement, during which she dropped the body. Following the loss, she emitted regular loud calls
sounding similar to lost calls, but these calls sounded slightly different to the observer, and RUB
looked very distressed. These calls were heard for several hours, progressively decreasing in
frequency and eventually stopped.

**Case 2: TRO's neonate 2006:** Neonate infant presumably died in its first day of life of unknown cause.

Not carried by mother.

*Infant sex and age:* The infant was born approx. ten days to the estimated term date, male

*Maternal rank, age and parity:* Low-ranking, middle-aged, multiparous.

*Case notes:* The infant was born during the night or on the previous day and it was first observed
early in the morning. The infant looked weak as he was unable to hold on with his back legs, and was
hanging underneath his mother by his arms, with his back legs hanging in the air, occasionally banging
against rocks. Every so often, TRO would use her arm to support the infant and walk on three legs.
The infant was not observed reaching out to the nipple and the mother was not observed trying to
bring him up to her nipple that morning. She remained peripheral from the group and looked quite

anxious about observers' presence, ~~who had to remain distant. Around mid-morning,~~ the infant
looked very weak, and the mother left it lying on the ground while she foraged within 1 m. TRO's
older daughter, 3-year old SCA, ~~came over to~~ the infant and held him, and he responded weakly. TRO
~~came back when~~ hearing her infant's reaction, and ~~her daughter, SCA, proceeded to groom~~ TRO. TRO
~~sat for her daughter,~~ and then ~~lay~~ down on top of the infant. The infant made no noise, and was
flattened into the sand. The mother then moved off to forage again, and ~~again her daughter went~~
to pick up her brother. The infant made a weak cry, and the mother came hurrying back, ~~collected~~
the infant up, and moved out of sight. ~~This was the last time that the infant was seen, alive or dead. No~~
~~sign of carrying~~ was recorded later that day or in the following day, despite the troop being followed.

**Case 3: SUL's neonate, 2006:** Death of unknown cause, not carried by the mother

*Infant sex and age:* Unknown sex, 1 day old

*Maternal rank, age and parity:* Low-ranking, middle-aged, likely primiparous (no live offspring)

*Case notes:* The infant was born around term, and was first observed alive ~~early in the morning,~~
gripping its mother ~~formally.~~ When first observed with her infant, the mother was associating with
the dominant male, ~~who was possibly the father.~~ ~~The baby was then not observed during most of~~
~~that day; when~~ next observed later that afternoon (around 5 pm), the baby ~~was not with the mother~~
~~anymore but~~ was in the possession of a ~~young low-ranking female,~~ SCA, ~~who was not related to the~~
~~infant.~~ It is notable that this young female had lost her neonate brother 7 days ~~before the date of this~~
~~observation,~~ and had been observed interacting with her dying brother (see previous Case Report).
The mother was sitting >20 m away ~~from her baby,~~ and did not seem concerned by SCA's possession
of the infant. SCA appeared ~~very~~ agitated, moving around ~~a lot,~~ and aggressively protecting the infant
from other inquisitive baboons. At this stage, the infant blinked once, and ~~also~~ convulsed ~~very~~ slightly,
but otherwise looked dead. The young female continued to hold the baby, and looked ~~very~~ distressed
when others ~~came around her to investigate the baby.~~ There were no signs of life by the point that
the baboons went onto the sleeping cliff, and the mother was ~~not observed around~~ her baby
anymore. SCA continued to carry the dead infant ~~all the following day.~~ In the morning she held it
~~against her~~ and was very careful with it. SCA ~~was observed holding~~ the baby's face up to hers and
~~making~~ muzzle to muzzle contact, most likely sniffing. The level of care and attention diminished over
the course of the day although she always kept the baby with her. By the afternoon SCA ~~was placing~~
it on the ground as she foraged, and probably left the corpse on the sleeping cliff that ~~following~~ night
because she was not observed with it the following day.

**Case 4: GAB's infant, 2006:** Death of unknown cause, though the infant showed delayed
development, carried for 3-4 days

*Infant sex and age:* Male, approx. 8 months old

*Maternal rank, age and parity:* Mid-ranking, young, multiparous (though no surviving offspring)

*Case notes:* The infant showed ~~an important~~ locomotor and physical developmental delay: he was
very small for his age, ~~and~~ in relatively poor condition, and was carried ventrally for an unusually long
period, up to ~~five months old~~. This might suggest a malformation, or a severe nutritional deficit. ~~He~~
~~was born to a mother that~~ had lost a two-month-old infant the year before. The infant was observed
dead ~~one morning and~~ was carried by GAB for two full days. ~~Because~~ there were no observers with
the troop the day before the ~~first observation of death, it is possible that the corpse was carried for~~
~~three days~~. The mother generally carried the corpse in her mouth, occasionally dropping the body to
forage, and was occasionally observed grooming ~~the corpse~~. It was probably finally abandoned on a
sleeping cliff as she was not observed with it the third day after ~~the first record of death~~.

**Case 5: MBA's foetus 2009:** Stillborn foetus, carried for 2 days

*Infant sex and age:* Female, almost-term foetus.

*Maternal rank, age and parity:* Low-ranking, young, nulliparous female.

*Case notes:* The probable cause of the abortion was a 2 m fall from a tree after the female had been
chased there by the dominant male of her troop during an inter-troop interaction. MBA carried the
corpse for two days before "losing" ~~the corpse~~ whilst foraging early on the second morning. The loss
of the corpse appeared accidental—MBA searched the area in which she had been foraging making
loud barks that sounded like lost calls. She continued searching as the troop moved away, and
remained in the area until the troop was out of sight ~~and she left~~ to catch up with the troop. A natal
adult male, 8 y.o. RAB, remained near MBA while she searched the area, and travelled back to the
troop with her. MBA continued to make loud barks intermittently throughout ~~the rest of~~ that day.

**Case 6: SAL's neonate 2010:** Death from unknown cause the day after birth, carried for 7 days

*Infant sex and age:* Unknown sex, 0 days old

*Maternal rank, age and parity:* Low-ranking, middle-aged, multiparous female

*Case notes:* SAL carried the corpse for 7 days, frequently grooming it. Initially, ~~the corpse was carried~~
ventrally, but after several hours she carried it in one ~~or the other~~ hand whilst travel-foraging. When
travelling rapidly, she ~~would hold~~ the corpse ventrally ~~and travel tripodally~~. SAL allowed her 5 year-
60 240 old daughter, SHI, to handle and groom the body, and her 3 year-old son, CYS, to play with ~~the corpse~~

241 while SAL was being groomed by SHI 3-4 days after the infant's death. No other baboon was observed
to be allowed to touch the corpse, though others showed interest in it. SAL started placing the corpse
in her lap while she foraged, then on the ground beside her, before foraging further and further away
from the corpse over the week. Eventually the corpse was, presumably, dropped in the night at the
sleeping cliff, as she was observed taking it onto the cliff in the evening but leaving the cliff without
it the next morning.

**Case 7: BRA's infant 2013:** Death from probable disease, carried for 2 days before recovered by
observers

*Infant sex and age:* Male, 27-28 weeks old

*Maternal rank, age and parity:* Mid-ranking, middle-aged multiparous female.

*Case notes:* The infant died during the night after two weeks of deteriorating condition; he was
observed with diarrhoea and defecating a large, pink nematode. On the morning after the infant's
death, BRA foraged some distance from the corpse. The male she associated with, and probable
father of the infant, CHO sat near the corpse (Fig. 1) and groomed it briefly at least twice. When the
group started to travel faster on the morning of the infant's death, BRA and CHO remained at the
back of the troop in a seemingly agitated and confused state. BRA was carrying the corpse ventrally,
walking tripodally and paused regularly. She would rest and bark occasionally. CHO was following her
at the back of the troop and sat near BRA and the body. He was, uncharacteristically, very wary of
human observers and threatened us regularly when approached. Two observers made several
attempts over 1 h that morning to recover the corpse to perform a necropsy to identify the cause of
the death. However, BRA screamed in protest and retrieved the corpse in the direction of the corpse,
and CHO charged the observers several times. It was not clear whether CHO's
aggressive behaviour towards the observers was motivated by BRA's screams, the proximity of the
observers to the corpse, or a combination of both. BRA carried the corpse for 2 days before an
observer retrieved it to perform the necropsy and take a biopsy, as at this stage she was regularly
foraging away from the corpse and being less vigilant towards it.

Figure 1. Case 7: ~~CHO's response to~~ BRA's infant's corpse

Caption: Adult male CHO sits near the corpse of BRA's infant as she forages nearby.

Case 8: PRE's infant 2013: Death due to illness

Infant sex and age: Female, 9-months old

Maternal rank, age and parity: Middle-ranking, middle-aged, multiparous female

Case notes: Six days before the death, observers recorded a visible deterioration of condition of the infant: some yellow vomit was seen around her mouth, she was very weak and lying in a lethargic state on the ground. She was being carried ventrally by her mother—something quite unusual for an infant of her age. The body could be inspected closely by one observer the day of the death (but not retrieved) and did not present any open injury. PRE was in a very agitated and nervous state during the days following the death, and was protective of the corpse. She groomed the corpse frequently and remained close to the dominant male VOL. It is unknown for how long PRE carried the corpse.

Case 9: GAB's infant 2013: Death likely due to infanticide, carried for 10 days.

Infant sex and age: Male, 7 months old

Maternal rank, age and parity: Mid-ranking, older, multiparous female

Case notes: The infant's corpse had a large cut on the right side of his body (from which internal organs were visible); GAB also had a 10-15 cm open, bleeding cut on the right cheek. Together, these injuries were strongly suggestive of infanticide. The first day, GAB removed and licked the organs of the corpse, but did not consume them. GAB carried the body that day, remaining at the edge of the group and actively protecting the corpse against conspecifics that approached to investigate the body. At the beginning of the day, GAB left the corpse on the ground to forage briefly. At the end of the day she would leave the corpse for longer, but always stay vigilant about the approach of other baboons or of the observers. She was seen resting and in close proximity with CHO, an adult immigrant male, in the middle of the day; and with KIL, an unrelated natal male, toward the end of the day. The day following the death, one observer was charged by an immigrant adult male, PLA, GAB's "friend" and the probable father of the infant, as she took pictures of the corpse. PLA continued to threaten the observer until she moved >30 m away from the corpse. The body deteriorated rapidly in the days following the death. By the 8th to 9th days, the corpse was desiccated, with only the skin and bones remaining. GAB carried the corpse for 10 days; on the tenth day, she started swelling again and would leave the body when foraging, ranging farther and farther away. However, GAB remained vigilant about the location of the corpse and would retrieve it when an observer approached. Late on the 10th day, she was observed without the corpse.

Case 10: MYR's early-term foetus 2013: Miscarriage of early-term foetus following alpha male replacement by an immigrant male

Infant sex and age: Unknown, foetus approx. 8 weeks.

Maternal rank, age and parity: High-ranking, older, multiparous

Case notes: MYR was pregnant with the offspring of the former dominant male, PNO. PNO had recently lost his dominant position to LIM, who had recently immigrated into the troop. LIM had been particularly aggressive toward MYR, the dominant female, since his arrival, attacking her repeatedly in very violent bouts in the weeks following his arrival. On the day of this observation, MYR was observed with vaginal bleeding. After bleeding for part of the day, a large "clot" was passed. MYR looked back at this stage, and manually removed the foetus, dropped it on the ground and walked away. LIM, who was following closely behind MYR, walked up to the foetus, and bent down to smell

it for a long time. We strongly suspected a case of sexually-selected foeticide. The ~~small~~ foetus was
collected by the observer and was approx. 10-15 cm long.

**Case 11: BRO's infant 2015:** Death from probable infanticide, carried for approx. 1 week

*Infant sex and age:* Female, approx. 12-13 weeks old.

*Maternal rank, age and parity:* High-ranking, young, primiparous female.

*Case notes:* The infant died from injuries that we strongly suspect were inflicted in an infanticide: ~~the~~

~~infant had~~ canine puncture wounds visible in the left arm and above the hips (visible in Fig. 2a, c).

BRO carried the corpse for approx. one week, grooming it frequently (Fig. 2a) and carrying it ventrally

whilst travelling (Fig. 2e) and foraging (Fig. 2b, c) and after desiccation. BRO inspected closely the

corpses' puncture wounds, often smelling her probing finger after doing so (Fig. 2f), and eventually

pulling part of the digestive system out of one wound above the left hip as a result of tugging on it

during grooming (visible in Fig. 2c). She also cleaned debris from inside the mouth of the infant. BRO

was associated with an adult male, MAR, who was the likely father of the infant (Fig. 2d). On one

occasion, MAR threatened an observer when she unknowingly approached the corpse that BRO had

left several meters away. ~~The observer at first did not understand why MAR threatened her as she~~

~~was approx. 7 m from the subgroup, a distance of observation that does not usually affect the~~

~~baboons' behaviour;~~ BRO continued foraging nearby and was not alarmed by the observer's

proximity to the subgroup or the corpse. This was not uncharacteristic, as BRO was generally very

relaxed around human observers. ~~As a recent immigrant to the troop, MAR may have been less~~

~~tolerant of the observer's proximity to the corpse; on noticing the corpse after a few minutes, the~~

~~observer moved away, MAR stopped threatening and he appeared to relax. Although MAR was not~~

~~observed interacting with the corpse, the threats may be indicative of a form of paternal~~

~~"protection."~~ BRO allowed at least one unidentifiable juvenile female (approx. 1 year old) approach

and interact with the corpse several days after her death.

Figure 2. Case 11, BRO's responses to her infant's death.

Caption: (a) BRO grooms the corpse. Note the puncture wounds visible in the arm and hip. (b,c) BRO forages with the corpse. Note that the digestive tract of the infant has been pulled out of the

342 puncture wound above the hip. (d) BRO grooms MAR, the probable father of the infant, leaving the
343 corpse nearby. (e) BRO carries the corpse ventrally. (f) BRO smells her finger after probing the
344 corpse's wounds.

**Case 12: BIL's infant 2017:** Death most likely from dehydration, two days after the death of her
mother, BIL.

*Infant sex and age:* Female, approx. 9-10 weeks old

*Maternal rank, age and parity:* Low-ranking, older, multiparous

*Case notes:* The cause of BIL's death is unknown, but the infant, who was still dependent, survived
her mother's death and remained with the troop. ~~the infant~~ was observed being carried and
groomed by juvenile baboons, particularly juvenile females and an unrelated, ~~7 year old juvenile~~
~~male~~, CYS, during the two days she survived her mother's death. The infant ~~would attempt~~ to forage
in close proximity to these individuals. ~~the infant's~~ adult older sister, BUB, had a ~~similarly aged~~
~~and was not observed approaching or carrying~~ her sister; the infant's juvenile older sister, 4 year-old
AMN, did attempt to approach ~~the infant~~, but was often supplanted by higher-ranking individuals
from different matriline. On the afternoon of the day the infant died, the corpse was ~~observed being~~
carried by CYS. This male was not previously associated closely with the infant or her mother. One
observer, AC, made detailed notes on this male's and others' behaviour towards the ~~infant's~~ corpse
over the following day (Box 1).

**Box 1:** Detailed observations of troop-members' responses to the corpse of an infant baboon ~~whose~~
~~mother was absent (i.e. died two days before her infant).~~

*15 July 2017*

1720: CYS ~~is observed carrying~~ corpse in his mouth by its tail.

1740: CYS carries the corpse by holding it to his ventrum up and onto the sleeping cliff.

*16 July 2017*

0610: CYS carries the corpse on his ventrum; he is one of the last individuals to leave the sleeping cliff
(Fig. 3a).

0618: CYS grooms the corpse (Fig. 3b).

0630: CYS grooms the corpse.

0645: CYS carries the corpse by the tail after the troop alarms at farmers (Fig. 3c).

0635: CYS approaches an adult male (unidentified) whilst carrying the corpse ventrally.

0649: Whilst remaining on the periphery of the group (Fig. 3d), CYS grooms the corpse, not foraging (Fig. 3e).

0656: CYS grooms the corpse, then carries the corpse very “gently” by the tail.

0721: CYS starts foraging whilst carrying the corpse; he is on the periphery of the troop with RUB, a high-ranking adult female.

0741: CYS receives grunts from NEC, a 7-year-old ~~juvenile~~ male, who approaches; CYS responds by tail-flagging and retreating, carrying the corpse by the tail.

0754: CYS “carries” the corpse by dragging it by the tail along the ground.

0805: Whilst foraging in a large *Salvadora persica* bush, CYS threatens ASP, a 5-year-old ~~juvenile~~ male, who approached the corpse without having seen it, as CYS had left it on the ground outside the bush while he foraged in the bush.

0805: CYS threatens ANA, CYS’s 4 year-old ~~juvenile~~ female sister, who approached the corpse. CYS then returns to the ground to gather the corpse, and carries it into the *Salvadora* where he forages with it in his lap (Fig. 3f).

0901: CYS carries the corpse by its right hand in his mouth. The troop alarms several times; each time CYS grooms the infant after the alarm, in a similar fashion to the grooming elicited between pairs after a troop alarm.

0926: CYS climbs into a *Prosopis glandulosa* tree with the infant by holding it to his ventrum and in his mouth by the tail.

1000: CYS threatens ULC, a higher-ranking, 7-year-old, adult female, unrelated to either CYS or the infant, for access to the corpse.

1005: CYS grooms the corpse.

1025: CYS threatens AMN, the 4-year-old ~~juvenile~~ sister of the dead ~~infant~~, for approaching the corpse, where he had left it on the ground approx. 8 m away. He takes the ~~infant~~ into the tree.

1054: BRU, an unrelated 7 year-old subadult male, approaches the corpse; CYS picks up the corpse to approach BRU whilst lip-smacking and making a come-hither face, an affiliative gesture (Fig. 3g). BRU returns the gestures and looks at the corpse.

1109: CYS grooms the corpse.

1143: The corpse is found by ANA and groomed, and she cleans ~~the corpse’s~~ mouth, removing some debris and inspecting it (Fig. 4a, b). ANA is approached by CYS, but not supplanted, and by AMN.

[revised manuscript text omitted]

DISCUSSION

There are few generalities about chacma baboons' responses to dead infants ~~to be made~~ from the 12 case reports that we present here. Although we observed some variation in the duration that mothers carried their infants' corpses, ~~in the majority of cases the mother carried her infant for more than one day (i.e. 8 of 11 cases, note that in one case (Case 12) the mother died before her infant and was therefore unable to carry it). However, as in other studies, there was variation in the duration (2-10 days) that the corpses, when carried, were carried. This variation does not indicate any clear pattern with the mother's age or parity, the cause of the infant's death, or the duration before the mother resumed cycling, though our sample is too small to assess these relationships statistically. In general, our anecdotes are similarly variable to those reporting many (>10) cases elsewhere [i.e. 9,10], with no obvious relationship between any of the recorded variables and whether a corpse is carried or not, or the carrying duration. In our discussion, we attempt to highlight some similarities and differences in chacma baboons' responses to dead infants in comparison to reports in other primates. We start by briefly assessing what support our observations can lend to some current hypotheses proposed to explain why primates carry dead infants before considering some of the "bigger" questions around the death of conspecifics, and what our anecdotes may suggest about the minds of baboons.~~

~~With respect to extrinsic influences on corpse carrying behaviour, the climate hypothesis suggests that prolonged carrying of infants' corpses is enabled by high altitude or dry climates because the infant's corpse decays less rapidly [reviewed in 3,10]. Tsaobis' climate is dry and corpses desiccate rapidly, meaning that our study population represents an interesting comparative point in assessing support for this hypothesis, under which we expect prolonged corpse carriage. However, in contrast to reports on chimpanzees (*Pan troglodytes*) [12], gelada monkeys (*Theropithecus gelada*) [10], and Japanese macaques (*Macaca fuscata*) [9], prolonged carriage on the order of weeks months has not yet been observed. Carrying duration in our sample can be long (10 d), but is in line with average, not extreme, carrying durations observed in other studies with multiple case reports from cold or dry climates [9,10]. As a result, this study does not support the climate hypothesis. However, this does not mean that climate does not play a role; rather, it suggests that there are other environmental factors that also need to be taken into account, for example, the weight of the corpse and the average daily distance covered by the group. At Tsaobis, the average day range is ~6.0 km [16], making it relatively costlier to transport a corpse in comparison to the shorter day ranges of~~

474 other wild species. For example, long carry durations have been reported for geladas [>43 days: ,10],
chimpanzees [up to 68 days: 12], provisioned Japanese macaques [17 days: ,9], and mountain gorillas
(*Gorilla beringei beringei*) [up to 27 days: 17] and all of these species have comparatively lower day
ranges, usually much lower than 5 km: geladas [between 1.3 and 4.1 km on average: 18–20],
chimpanzees [2-5 km: 21], the provisioned Takasakiyama macaques [0.7-2.5 km: 22] and mountain
gorillas [0.2-1.7 km: 23]. On this basis, it may be a combination of external factors that affect corpse
carrying duration. These might include constraints set by both the daily distance of the travel group,
with longer distances imposing a greater physical cost of exhaustion on the mother, and climate, with
wetter climates associated with the more rapid decomposition of the corpse.

Do baboons realise that dead infants are dead? In general, it is difficult to determine what
would constitute behavioural evidence of an awareness of death and it may be easier to identify
behavioural evidence of unawareness. Some have argued that the treatment of the corpse in a
manner reminiscent of its treatment in life could indicate unawareness [reviewed in 3]. In support of
this, most baboon mothers, and even unrelated individuals, were observed grooming the infants'
corpses and carrying the corpses ventrally. Likewise, two juveniles who interacted with BIL's infant's
corpse (Case 12) "used" it in a similar social-buffering manner to that of live infants that are used
during social interactions. Similarly, males' "use" of infants during social interactions is common in
Barbary macaques (*Macaca sylvanus*) and males of all ages in this species have been observed to use
infants' corpses in much the same way as live infants [24]. The examples of the corpse being used as
a social buffer in Case 12 could indicate that the baboons were treating the corpse as they would in
life because they were unaware that it was dead. Conversely, treatment of the corpse in a manner
different to its treatment in life could indicate an awareness that the infant was dead [as suggested
for chimpanzees: 12; see also: 3]. Our reports provide some support of this observation. For example,
baboons at our site have never been observed to carry live infants in one hand along the ground, or
in the mouth, even when the infant is sick or lethargic and unable to cling to the mother.
Furthermore, in two cases we observed individuals cleaning inside the mouth of the corpse (Cases
11, 12), which has never been observed being done to live individuals (discussed in greater detail
below).

Does the social bond with a dead individual, either through kinship or affiliation, predict the
survivor's response to the deceased? Given that we report only individuals' responses to infants'
deaths, rather than non infants' deaths, our contribution to such a question is limited as in most

cases the responding individual was the mother, who was the most closely-bonded to the infant.
However, in ~~three of the 12 reported cases~~ (Cases 2, 3 and 12), individuals other than the mother
interacted extensively with the corpse soon after death. In two of those cases, the infant died shortly
after birth, the mother was still alive and the ~~infants' corpses were~~ handled by a third individual, who
provided greater ~~posthumous~~ care than the mother. It has been argued ~~in similar cases that because~~
~~the infants died soon after birth the mother did not have time to form a strong bond, which could~~
~~have resulted in her decreased response~~ to her infant [8]. However, our data contradict this particular
interpretation of the social-bonds hypothesis: one mother showed extensive carrying of her
~~neonates' corpse~~ (Case 6), and two stillborn fetuses were also carried for several days and both
**mothers seemed distressed on prematurely losing the corpses** (Cases 1 and 5). The third case where
the infant's corpse was handled mostly by unrelated individuals (Case 12) lends some ~~speculative~~
support to the social-bonds hypothesis. ~~In this case, the individuals observed interacting with the~~
corpse of an already-deceased mother's infant behaved ~~in a similar manner to that of mothers in~~
~~response to~~ their own infants' corpses: they carried and groomed the body, foraged with it nearby,
and ~~were protective of the corpse on others' approaches to it~~. However, ~~while it could be argued~~
~~that this behaviour contradicts the social bonds hypothesis (because un-bonded individuals were~~
~~responding to the corpse)~~, the individuals involved performed these behaviours for less than a day
~~whereas only mothers have been observed to~~ carry their own infants' corpses for long periods (up to
10 days). ~~It thus seems that,~~ even when given the "opportunity" (through the absence of the mother)
to carry an infant's corpse for long periods, this **is** not done by non-maternal individuals in chacma
baboons. ~~This is similar to reports in geladas, where~~ non-maternal individuals have been observed
carrying infant corpses for a day or two but no longer [10]. However, we note that some mothers
also carried their own infants' corpses for less than a day—comparable to ~~the investment shown by~~
~~the~~ unrelated individuals. This **is** further highlights the multifactorial nature of this behaviour.

Observations of other ~~non-maternal individuals' behaviour~~ lend further support to the social-
bonds hypothesis. ~~we report three cases (7, 9, 11) where an~~ adult male—the probable father of the
dead infant in all cases—provided forms of ~~posthumous~~ "protection." ~~In all cases, the males~~
~~performed a type of "guarding" behaviour by sitting near the corpse when the mother temporarily~~
moved away, or by threatening ~~an observer(s) when the corpse was (sometimes accidentally)~~
~~approached~~. Such guarding behaviour of an infant's corpse by an adult male has previously been
reported in wild chacma baboons [25], ~~though in all cases reported here we can confirm that the~~

 ~~guarding male was closely associated with the mother and her infant, lending support to the social~~
 ~~bonds hypothesis.~~ Notably, during one guarding event, CHO briefly groomed the infant's corpse he
 ~~was guarding.~~ To our knowledge, such paternal protection towards an infant's corpse has not yet
 been reported in any primate ~~reviewed in 3], though we note that (presumably) unrelated male~~
 Barbary macaques (*Macaca sylvestrus*) regularly groom infants' corpses as they do with live infants
 [24]. However, in the macaque case, the behaviour is not necessarily protective, as the infants were
 used as social buffers that ~~benefitted~~ the carrying males. Paternal corpse grooming is perhaps not
 surprising for chacma baboons given males' investment in particular infants during life [26,27] and
 ~~the previously documented corpse guarding behaviour of an adult male towards a dead infant [25]~~
 ~~and documented corpse carrying in an olive baboon male (*P. anubis*) [28], indicating male baboons'~~
 ~~involvement in corpse "protection."~~ However, it is perhaps surprising that these behaviours have not
 ~~been documented~~ in other primate species with paternal access to offspring i.e. in other stable
 group-living primates with resident fathers, such as geladas, macaques and snub-nosed monkeys.
 Overall these observations lend further support to the social-bonds hypothesis.

We report an early-term miscarriage that was likely a "Bruce effect"-induced abortion (Case
 10) and two stillbirths (Cases 1 & 5). ~~What is notable about the miscarriage is the mother's lack of~~
 ~~response to the terminated foetus in comparison to the mothers' responses elicited in the stillbirths.~~
 A similar observation has also been made in Yunnan snub-nosed monkeys (*Rhinopithecus bieti*), with
 an aborted foetus being abandoned and not carried ~~(the mother could not be determined)~~ and a
 stillborn foetus being carried for one day [29]. Although ~~this is just two cases and it is important to~~
 ~~avoid over-interpretation,~~ these observations ~~do~~ lend some support to two of the proposed
 hypotheses for maternal corpse-carrying behaviour. In the first case, ~~MYR and the unidentified snub-~~
 ~~nosed monkey mother may not have carried the miscarried foetus because it lacked the cues~~
 ~~associated with an infant that the stillbirth foetuses had,~~ supporting the *infant-cues hypothesis*.
 Alternatively ~~(or additionally), in the miscarriage, hormones and neuropeptides that are~~
 hypothesised to trigger maternal ~~carrying~~ behaviour [30] may have been too low to elicit carrying
 behaviour (or any interest in the foetus) due to the early stages of the **pregnancies**, lending support
 to the *hormonal hypothesis*.

The stillbirth observations could shed light on which pregnancy ~~and/or~~ parturition hormones
 may be the most likely ~~candidates that trigger~~ maternal carrying behaviour in very early infant deaths.
 For example, stillbirths are associated with lower prepartum urinary oestrones in captive hamadryas

baboons (*P. anubis*) [31] and lower faecal oestrogens in the trimester of the pregnancy loss in wild
yellow baboons (*P. cynocephalus*) [32]. In both species, progesterone levels are comparable in
females with live and stillbirths [31,32], which may suggest that progesterone, but not oestrogen, is
key in facilitating carrying behaviour after birth. This is supported by hormonal observations in low-
caring and aberrant baboon mothering behaviour: hamadryas mothers who maintained less contact
with their infants had higher prepartum and lower postpartum urinary metabolites of progesterone
[33], suggesting that “normal” levels of progesterone may encourage contact between a mother and
her offspring, which may carry-over to stillborn infants. Similarly, postpartum behavioural difficulties
in hamadryas baboons were associated with elevated oestrone:progesterone ratios, again
implicating (relatively) low progesterone in reduced infant care [31]. However, hormones cannot be
implicated in maternal corpse-carrying behaviour of older infants because (1) ovarian hormone levels
rapidly return to pre-pregnancy levels in primates, for example, in wild yellow baboons [32,34], and
(2) post-puerperium maternal behaviour is maintained not by hormones but by behavioural
interactions between the mother and the infant [30]. It is possible that variation in other hormones,
such as oxytocin, may be responsible for maternal corpse care post-puerperium period [28];
however, the data to test this hypothesis are currently lacking.

Finally, we report a rare thanatological behaviour, namely the mouth and wound cleaning
of the deceased individual (Cases 11, 12). Two individuals were observed cleaning inside the mouths of
(different) infants’ corpses, taking care to remove and inspect debris that had become lodged in the
mouths, and one of these two individuals also cleaned a wound on her infant’s corpse. Similar
behaviours have been reported recently, including the probing of dead infants’ corpses by a captive
Tonkean macaque mother (*Macaca tonkeana*) [35] and a human commensal living juvenile female
bonnet monkey (*M. radiata*), who was not the mother of the infant whose mouth she investigated
[36]. In another case, this behaviour has been discussed as evidence for compassion towards the
dead individual, specifically, posthumous teeth cleaning behaviour of an adolescent male
chimpanzee’s corpse [7]. Evidence for compassion has also been hypothesised in a Sichuan snub-
nosed monkey’s (*Rhinopithecus roxellana*) behaviour towards a recently-deceased individual, which
was described as “exploration, attempts to elicit a response from the [deceased individual], and
affiliative acts including embracing” [37].

Does mouth- and wound-cleaning behaviour suggest that individuals, in our case chacma
baboons, are “compassionate” towards dead individuals? We would suggest that such observational

~~reports provide only limited insight into the question of compassion~~ for two reasons. First, ~~it is not~~
~~clear what would constitute evidence of compassion towards the dead. No clear definition of animal~~
compassion has been proposed by these previous studies, but compassion in humans is defined as
“sympathetic pity and concern for the sufferings or misfortunes of others” ~~and as such~~ cannot be
directly observed and must be inferred indirectly through ~~its potential~~ behavioural manifestations. ~~It~~
~~may be that~~ the described behaviours are evidence of ~~underlying~~ compassion, but the provided
justifications are tenuous. In the first case, the justification was that the mouth-cleaning behaviour
was “prolonged and detailed” (p. 921, the behaviour lasted approx. 2 minutes); in both cases the
authors cite the fact that the behaviour was performed by an individual who had a close social bond
to the deceased individual as evidence of compassion [7,37]. While the perception of an experienced
observer is ~~very useful~~ when trying to interpret a complex behaviour ~~we would even argue that~~
~~observers’ perceptions of their subjects’ emotional state cannot be replaced by detailed qualitative~~
~~or quantitative reports – it remains~~ unclear why such behaviour would be more compelling than a
long grooming bout ~~or~~ a different body location, for example, which is frequent in the primate
thanatological literature [reviewed in 3]. The duration of such behaviour could simply reflect the fact
that the corpse is unresponsive. In our study, affiliative acts (grooming) were performed by most
individuals who interacted with the corpses, not only those that were closely bonded, and an
unrelated, unbonded individual was observed cleaning the mouth of a dead infant, indicating that
this unusual behaviour is not restricted to closely-bonded individuals.

Second, no alternative explanations ~~than~~ compassion have been proposed to explain these
atypical behaviours, which could, for instance, equally indicate grief management responses in the
bereaved individuals, or curiosity ~~towards~~ death in the non-bereaved individuals. What is clear is that
teeth- and mouth-cleaning behaviours have never been observed being directed towards live
individuals, and that in itself makes them valuable to describe and discuss. Empirical attempts to
decipher the nature of the mental and emotional processes underlying such observations will
undoubtedly be very challenging, and may be best addressed by establishing systematic data
collection protocols around dead and ~~heavily~~ injured individuals in order to support some level of
hypothesis testing, as well as to document the apparent emotional state of individuals involved in
~~such interpretations, as this will be crucial for their interpretation.~~

We add to the growing body of literature on primate thanatology, and show how new data
can generate both ~~more~~ support for existing hypotheses and more questions to address. Although

quantitative ~~data on individuals' behavioural~~ responses towards infants' corpses should ~~now~~ be
targeted to move the field forward [8] we hope that we have demonstrated that systematic
publication of "detailed anecdotes" [see also: 13] can be ~~useful and informative to further our~~
understanding of some of the "big" thanatological questions, especially ~~since the scientific value of~~
~~such reports will increase as cases~~ accumulate from a variety of sites and species. We therefore
encourage future data **collection** in this detailed qualitative format ~~from existing field sites~~ as well as,
if possible, ~~the~~ implementation of specific behavioural data protocols ~~on individuals' deaths, which~~
~~will be essential to future hypothesis testing.~~

ACKNOWLEDGEMENTS

We thank the Tsaobis Baboon Project field teams from 2006-2017 for their dedication and patience
every year that made following the baboons possible. We are grateful to the Ministry of Lands and
Resettlement for permission to work at Tsaobis Leopard Park, the Gobabeb Training and Research
Centre for affiliation, and the Ministry of Environment and Tourism for research permission in
Namibia. We are also grateful to the Snyman and Wittreich families for permission to work on
neighbouring farms. This paper is a publication of the ZSL Institute of Zoology's Tsaobis Baboon
Project.

FUNDING

The authors are grateful to the following funders who supported their research at Tsaobis: the Leakey
Foundation (2009-2010; 2016-2017), the Animal Behavior Society (USA) (2009), the International
Primatological Society (2009), the Explorers Club Exploration Fund (2009); Primate Society of Great
Britain (2013); the Gibbs Travelling Research Fellowship, Newnham College (2015); Fondation des
Treilles (2017); and Agence Nationale de la Recherche Labex IAST (2017).

REFERENCES

- 1. Anderson JR, Biro D, Pettitt P. 2018 Evolutionary thanatology. *Philos. Trans. R. Soc. B Biol. Sci.* **373**.
(doi:doi.org/10.1098/rstb.2017.0262)
- 2. Anderson JR. 2016 Comparative thanatology. *Curr. Biol.* **26**, R553–R556.
- 3. Gonçalves A, Carvalho S. 2019 Death among primates: a critical review of non-human primate
interactions towards their dead and dying. *Biol. Rev.*
- 4. Piel AK, Stewart FA. 2015 Non-Human Animal Responses towards the Dead and Death: A Comparative
Approach to Understanding the Evolution of Human Mortuary Practices. In *Death Rituals, Social Order
and the Archaeology of Immortality in the Ancient World: 'Death Shall Have No Dominion'* (eds C
Renfrew, MJ Boyd, I Morley), pp. 15–26. Cambridge University Press.
(doi:10.1017/CBO9781316014509.003)
- 5. Stewart FA, Piel AK, O'Malley RC. 2012 Responses of chimpanzees to a recently dead community
member at Gombe National Park, Tanzania. *Am. J. Primatol.* **74**, 1–7.
- 6. Buhl JS, Aure B, Ruiz-Lambides A, Gonzalez-Martinez J, Platt ML, Brent LJN. 2012 Response of Rhesus
Macaques (*Macaca mulatta*) to the Body of a Group Member That Died from a Fatal Attack. *Int. J.
Primatol.* **33**, 860–871. (doi:10.1007/s10764-012-9624-1)
- 7. van Leeuwen EJ, Mulenga IC, Bodamer MD, Cronin KA. 2016 Chimpanzees' responses to the dead body
of a 9-year-old group member. *Am. J. Primatol.* **78**, 914–922.
- 8. Watson CF, Matsuzawa T. 2018 Behaviour of nonhuman primate mothers toward their dead infants:
uncovering mechanisms. *Philos. Trans. R. Soc. B Biol. Sci.* **373**, 20170261.
- 9. Sugiyama Y, Kurita H, Matsumi T, Kimoto S, Shimomura T. 2009 Carrying of dead infants by Japanese
macaque (*Macaca fuscata*) and others. *Anthropol. Sci.* **117**, 113–119.
- 10. Fashing PJ *et al.* 2011 Death among geladas (*Theropithecus gelada*): a broader perspective on
mummified infants and primate thanatology. *Am. J. Primatol.* **73**, 405–409.
- 11. Masi S. 2019 Reaction to allospecific death and to an unanimated gorilla infant in wild western gorillas:
insights into death recognition and prolonged maternal carrying. *Primates*, 1–10.
- 12. Biro D, Humle T, Koops K, Sousa C, Hayashi M, Matsuzawa T. 2010 Chimpanzee mothers at Bossou,
Guinea carry the mummified remains of their dead infants. *Curr. Biol.* **20**, R351–R352.
(doi:https://doi.org/10.1016/j.cub.2010.02.031)
- 13. Campbell LAD, Tkaczynski PJ, Mouna M, Qarro M, Waterman AD, Majolo B. 2016 Behavioral responses to
injury and death in wild Barbary macaques (*Macaca sylvanus*). *Primates* **57**, 309–315.
(doi:10.1007/s10329-016-0540-4)
- 14. Marshall HH, Carter AJ, Ashford A, Rowcliffe JM, Cowlshaw G. 2013 How do foragers decide when to
leave a patch? A test of alternative models under natural and experimental conditions. *J. Anim. Ecol.* **82**,
894–902.
- 15. Carter AJ, Marshall HH, Heinsohn R, Cowlshaw G. 2013 Personality predicts decision making only when
information is unreliable. *Anim. Behav.* **86**, 633–639.

16. Benavides JA, Huchard E, Pettorelli N, King AJ, Brown ME, Archer CE, Appleton CC, Raymond M,
Cowlshaw G. 2012 From parasite encounter to infection: Multiple-scale drivers of parasite richness in a
wild social primate population. *Am. J. Phys. Anthropol.* **147**, 52–63.
17. Warren Y, Williamson EA. 2004 Transport of dead infant mountain gorillas by mothers and unrelated
females. *Zoo Biol. Publ. Affil. Am. Zoo Aquar. Assoc.* **23**, 375–378.
18. Iwamoto T, Dunbar R. 1983 Thermoregulation, habitat quality and the behavioural ecology of gelada
baboons. *J. Anim. Ecol.* , 357–366.
19. Moua C. 2015 Long-term ranging patterns of wild gelada monkeys (*Theropithecus gelada*) on an intact
afro-alpine grassland at Guassa, Ethiopia. PhD Thesis, California State University, Fullerton.
20. Moges A. 2019 Population Status, Diets, Activity Budget and Range Use By Arsi Gelada (*Theropithecus*
*Gelada Arsi*) In Eastern Arsi, Ethiopia. PhD Thesis, Addis Ababa University.
21. Pontzer H, Wrangham RW. 2006 Ontogeny of ranging in wild chimpanzees. *Int. J. Primatol.* **27**, 295.
22. Tsuji Y. 2010 Regional, temporal, and interindividual variation in the feeding ecology of Japanese
macaques. In *The Japanese macaques*, pp. 99–127. Springer. 25
23. Ganas J, Robbins MM. 2005 Ranging behavior of the mountain gorillas (*Gorilla beringei beringei*) in
Bwindi Impenetrable National Park, Uganda: a test of the ecological constraints model. *Behav. Ecol.*
*Sociobiol.* **58**, 277–288. 29
24. Merz E. 1978 Male-male interactions with dead infants in *Macaca sylvanus*. *Primates* **19**, 749–754.
(doi:10.1007/BF02373640)
25. Cheney DL, Seyfarth RM. 2007 *Baboon Metaphysics: the Evolution of a Social Mind*. Chicago: University
of Chicago Press.
26. Huchard E, Alvergne A, Féjan D, Knapp LA, Cowlshaw G, Raymond M. 2010 More than friends?
Behavioural and genetic aspects of heterosexual associations in wild chacma baboons. *Behav. Ecol.*
*Sociobiol.* **64**, 769–781.
27. Huchard E, Charpentier MJ, Marshall H, King AJ, Knapp LA, Cowlshaw G. 2013 Paternal effects on access
to resources in a promiscuous primate society. *Behav. Ecol.* **24**, 229–236. (doi:10.1093/beheco/ars158)
28. Bercovitch FB. 2019 A comparative perspective on the evolution of mammalian reactions to dead
conspecifics. *Primates* , 1–8.
29. Li T, Ren B, Li D, Zhang Y, Li M. 2012 Maternal responses to dead infants in Yunnan snub-nosed monkey
(*Rhinopithecus bieti*) in the Baimaxueshan Nature Reserve, Yunnan, China. *Primates* **53**, 127–132.
(doi:10.1007/s10329-012-0293-7)
30. Rosenblatt J. 1994 Psychobiology of maternal behavior: contribution to the clinical understanding of
maternal behavior among humans. *Acta Paediatr.* **83**, 3–8.
31. French JA, Koban T, Rukstalis M, Ramirez SM, Bardi M, Brent L. 2004 Excretion of urinary steroids in pre-
and postpartum female baboons. *Gen. Comp. Endocrinol.* **137**, 69–77.
32. Beehner JC, Nguyen N, Wango EO, Alberts SC, Altmann J. 2006 The endocrinology of pregnancy and fetal
loss in wild baboons. *Horm. Behav.* **49**, 688–699.

33. Bardi M, French JA, Ramirez SM, Brent L. 2004 The role of the endocrine system in baboon maternal
behavior. *Biol. Psychiatry* **55**, 724–732.
34. Altmann J, Lynch JW, Nguyen N, Alberts SC, Gesquiere LR. 2004 Life-history correlates of steroid
concentrations in wild peripartum baboons. *Am. J. Primatol.* ~~*Off. J. Am. Soc. Primatol.*~~ **64**, 95–106.
35. De Marco A, Cozzolino R, Thierry B. 2018 Prolonged transport and cannibalism of mummified infant
remains by a Tonkean macaque mother. *Primates* **59**, 55–59.
36. Das S, Erinjery JJ, Desai N, Mohan K, Kumara HN, Singh M. 2019 Deceased-infant carrying in nonhuman
anthropoids: Insights from systematic analysis and case studies of bonnet macaques (*Macaca radiata*)
and lion-tailed macaques (*Macaca silenus*). *J. Comp. Psychol.* **133**, 156.
37. Yang B, Anderson JR, Li B-G. 2016 Tending a dying adult in a wild multi-level primate society. *Curr. Biol.*
**26**, R403–R404. (doi:<https://doi.org/10.1016/j.cub.2016.03.062>)

Case 7, CHO's response to BRA's infant
Adult male CHO sits near the corpse of BRA's infant as she forages nearby

1236x824mm (72 x 72 DPI)

Case 11, BRO's responses to her infant's death. Caption: (a) BRO grooms the corpse. Note the puncture wounds visible in the arm and hip. (b,c) BRO forages with the corpse. Note that the digestive tract of the infant has been pulled out of the puncture wound above the hip. (d) BRO grooms MAR, the probable father of the infant, leaving the corpse nearby. (e) BRO carries the corpse ventrally. (f) BRO smells her finger after probing the corpse's wounds.

231x287mm (72 x 72 DPI)

Case 12, CYS's responses to BIL's infant

Caption: (a) CYS carries the corpse ventrally from the sleeping cliff. (b) CYS grooms the corpse on leaving the sleeping cliff. (c) CYS carries the corpse in his mouth by the tail. (d) CYS remains on the periphery of the troop, not foraging and continuing to handle the corpse. (e) CYS continues to groom the corpse on the periphery of the troop. (f) CYS forages with the corpse after it was approached by other troop members. (g) Using the corpse in the manner of a social buffer, CYS approaches BRU, who performs a lip-smack at the corpse.

202x286mm (72 x 72 DPI)

Case 12, ANA's responses to BIL's infant

Caption: (a) ANA cleans inside and (b) removes debris from the mouth of the corpse. (c) ANA approaches BUB, BIL's adult daughter, who had a similarly-aged son at the time of BIL's infant's death.

202x286mm (72 x 72 DPI)

Appendix B

We thank the reviewers for their constructive feedback on our manuscript. Please find responses to the comments below in blue text.

Reviewer: 1

Comments to the Author(s)

As the authors state, it is important for more animal behaviour researchers to describe and publish carefully observed and described instances of responses to dead conspecifics, to advance the field of evolutionary/comparative thanatology. Here, the focus is on responses by mothers and other individuals to dead infant chacma baboons, a species that appears under-represented in the literature. The authors provide several nice descriptions of what happened to dead infants of varying ages and likely causes of death, and relate their observations to some hypotheses that exist in the literature. I will be happy to see this paper published.

RESPONSE: We thank the reviewer for this positive feedback.

I have no major criticisms of the work, but shortly after I started to read the manuscript I felt that it was over-written, with too many redundant phrases, repetitions, and a lack of concision. I'm attaching the manuscript with many suggested changes aimed at increasing conciseness and making the paper more readable to a scientific, rather than a popular readership. Some of the comments are questions, so the authors are advised to read them all carefully and take it from there.

RESPONSE: Again, thank you for your time with this. We have incorporated the suggested changes and addresses the questions raised.

Additionally, I offer some further comments aimed at making the paper both tighter and stronger:

The recently published paper by Das et al. deserves more than one passing citation, as it also addresses several hypotheses using a relatively large and multi-species data base.

RESPONSE: We have now added to the Introduction: "This hypothesis was not supported by a recent cross-species analysis of anthropoid primates (great apes, old world and new world monkeys) that controlled for phylogenetic relatedness. The mother's age, the cause of the infant's death and the degree of the species' arboreality determined the length of time a corpse was carried [11]" LL58-62.

The authors repeatedly surmise that an attending adult male was the likely or probably father of the dead infant. But without supporting evidence this is too speculative. The baboons are studied every winter, but doesn't that mean that the females conceived at a time when no observations were conducted, and therefore the father is simply unknown? If so, I would recommend dropping the suggestion each time, and somehow working into the Discussion a single sentence alluding to the fact that some of the attending adult males appeared to have long-term friendships with the mothers.

RESPONSE: Although the baboons are followed only 3-8 months per year, in many cases we know who consorted with the female in her conceptive cycle. However, since we have not yet analysed the microsatellite data in most cases, we now mention only that the male was the mother's friend at the time and that these friends are often the fathers of the infants in our population (LL107-110). For one infant, preliminary microsatellite analysis combined with behavioural data implicate one male as the genetic sire; he was also the mother's friend (LL257-262). We have also revised the discussion LL518-532.

Case 5: I'd like more information. When was the foetus expelled after the mother's fall?
RESPONSE: We have now added this information. LL227.

In several places the authors describe "protection" of the corpse when others approached. Again, some clarification would be helpful.... Was protection ever aggressive, or did the corpse-handler simply avoid approaches by others, for example moving away, or simply shielding the corpse with its body?

RESPONSE: We have now better described the protective behaviour we observed. LL285-287, 312-313.

Line 474: When discussing day ranges of species in relation to length of corpse-carrying, I'd recommend trying to get day range length information for the specific study sites where long durations have been reported, e.g., Bossou for chimpanzees.

RESPONSE: We have changed these values where we have found these data for these sites, and contacted a researcher at Bossou when we did not find these data published. LL474-476.

Line 619: I totally disagree that mouth-grooming and teeth-cleaning have "never" been observed between living individuals. I've watched stumptail macaques and tonkean macaques fold back the lips of a groomee and carefully groom the exposed area, and I suspect that somewhere in the hundreds of papers and chapters on grooming that this kind of behaviour is described (albeit not highlighted as being special; it may be simply included as face-directed grooming). Bill McGrew also published a paper entitled "chimpanzee dentistry," in which the interactants were very much alive! Again, mouth-grooming in chimps might well be much more common than the authors suggest.

RESPONSE: We apologise for our lack of clarity. We have now changed this to: "What is clear is that teeth- and mouth-cleaning behaviours have never been observed being directed towards live baboons at our field site, and that in itself makes them valuable to describe and discuss. (We note, however, that teeth-cleaning with tools has been described in chimpanzees [39].)" LL597-598.

I hope these comments and suggestions are useful.

RESPONSE: Yes, very much so. Thank you for taking the time to pass these on.

Reviewer: 2

Comments to the Author(s)

I think this article provides important implications to expand our understanding on the animals' responses to death. Existing literatures on primate thanatology have mainly come from forested areas with humid climate. This article presents unique data-set from a dried habitat of chacma baboons that travel long distances and use sleeping cliff at night. The comparison with other primates will highlight the characteristics in each clade; even in a dry habitat, the length of carrying in chacma baboons was not so much different from other monkey species.

RESPONSE: We thank the reviewer for these encouraging words.

Points to be considered in the revise is listed below.

P3 L11: The word “grieve” is too speculative. I suggest to remove this sentence from the abstract. Instead, I advise to include the description of “awareness” since the word is included in the key words but missing from the abstract text.

RESPONSE: We do not propose to answer the question about grief, but we do believe that it is possible to use these kinds of observations to inform this question. We have, however, removed its mention from the abstract, as suggested by the reviewer.

P6 L92: Put parenthesis to the species name, chacma baboons’ (*Papio ursinus*) responses

RESPONSE: Done.

P6 L104-: More descriptions on the study site and the subject troops should be included in this part. The climate is estimated to be dry, but authors should give concrete information such as annual rainfall and seasonality, if any (and add month to each case report). The composition of the two study troops should be included, even if it’s rough summary, such as xx-xx individuals for troop J (including x-x adult male and x-x adult female). Also, give basic information for behavioral characteristics of chacma baboons for readers other than primatologists; diet, daily travel distance, society, mating (and consort?).

RESPONSE: We have added the requested information. LL101-110; L115.

P6 L112: higher than the (previous?) reports suggests... The following long sentence was not so clear for me. Inclusion of the total number of observation days during the 13 years from 2000 would be informative to have rough ideas about the frequency of observing infants’ deaths.

RESPONSE: We were referring to the cases reported in this study. This sentence has been edited for clarity.

P7 L131-133: Number of cases for each category should be checked; $2+3+7+1=13$

RESPONSE: This has been changed.

P9 L187: When was the timing of confirming the mother without the infant? Some hours later on the same day or the next morning?

RESPONSE: We are more specific now in our revisions L190.

P10 L217: lost a two-month-old infant the year before (not included in this report due to the lack of precise records?)

RESPONSE: We did not collect data on all infants’ deaths, in part because not all corpses were carried. We hope the new edits (LL121-125) make this clearer.

P10 L227: MBA carried the corpse in which manner (ventrally or in hand/mouth)?

RESPONSE: We have now added this information. LL229, 233

P12 L279: Why the length of carrying was not recorded in this case? Even if the end day was not recorded, you may say that it was carried at least for x days.

RESPONSE: We now provide a minimum carry duration. L288

P13 L286: Please explain the possible cause of infanticide and frequency of happening in chacma baboons.

RESPONSE: We now provide information on the frequency of infanticide when describing baboon society. LL106-107

P13: Case 9 and Case 10 happened in the similar timing?

RESPONSE: We now report the month, as requested earlier. In this case, the events were months apart in different troops.

P13 L303: please add the note of “not carried (by the mother)”

RESPONSE: added.

P16-18: Box 1 may be better to be included as a supplementary material, leaving some important parts (such as social buffering between males in L397-) in the main text.

RESPONSE: As there are no page limits in RSOS because it is online-only, we prefer to keep this information in the main text. By including it in a Box, it is parenthetical to the text, and the Figures capture many of the main points should the reader choose to “skip” the Box.

P27: The part discussing about “compassion” may be too speculative. Please try to discuss your gained results in a scientific manner.

RESPONSE: We make the point about compassion to provide caution to others’ interpretations of primates’ responses to the dead and dying. For this reason, we prefer to keep this discussion. However, having taken the position that others’ interpretations may over-reach the data available, we do not agree that such a discussion is not done in “a scientific manner.” This is because evidence for compassion in free-ranging animals can, at the moment, come only from the observation of behaviour. We make this point more explicit in our revised manuscript. LL579-580

P28 L605: p. 921 of which reference?

RESPONSE: We have now added the reference here, too.